



# A novel European windstorm dataset based on ERA5 reanalysis from 1940 to present

Lorenzo Sangelantoni[1], Stefano Tibaldi[1], Leone Cavicchia[1], Enrico Scoccimarro[1], Pier Luigi Vidale[2], Kevin I. Hodges[2], Vivien Mavel[3], Mattia Almansi[3], Chiara Cagnazzo[4], and Samuel Almond[4]

[1] CMCC Foundation - Euro-Mediterranean Center on Climate Change, Bologna, Italy
[2] National Centre for Atmospheric Science, Dept. of Meteorology, University of Reading, Reading, UK
[3] B-Open Solutions srl, Rome, Italy
[4] ECMWF, Bonn, Germany

*Correspondence to*: Lorenzo Sangelantoni (lorenzo.sangelantoni@cmcc.it)

**Abstract.** In this work, we present and preliminarily evaluate a novel dataset of European windstorms associated with extratropical cyclones (ETCs) based on the whole ERA5 reanalysis period (1940-present). This dataset is produced within the Copernicus Climate Change Service (C3S) Enhanced Operational Windstorm Service (EWS), to promote a knowledge-based assessment of the nature and temporal evolution of European windstorms associated with ETC. Such a dataset is primarily
thought to provide high-quality, standardized data on windstorms that support various industries, particularly insurance and risk management, by offering insights into the intensity, density spatial patterns, and, if coupled downstream, with vulnerability and exposure information, the impact of windstorms. EWS includes two datasets: windstorm tracks, based on two tracking algorithms (TRACK and TempestExtremes), and windstorm footprints, produced considering both original-resolution ERA5 variables and statistically downscaled ERA5 variables, with a target grid at 1 km resolution. A preliminary analysis of the
datasets shows increasing number of cold-semester windstorms and the associated footprint wind gusts magnitude over a portion of the European territory. The choice of the tracking algorithm is shown to be an important factor in the decision-making process, as it results in non-negligible uncertainties in main windstorm statistics.

## 1 Introduction

Extratropical cyclones (ETCs) are dominant meteorological structures playing a crucial role in midlatitudes and are ubiquitous
sources of day-to-day weather variability. ETCs are important components of the atmospheric general circulation, transporting heat, moisture, and momentum (Catto, 2016; Degenhardt et al., 2023). At the same time, ETCs are also responsible for heavy precipitation events, strong surface winds and wind gusts (Ulbrich et al., 2009). Windstorms produced by severe ETCs can expose populations to hazards and cause widespread and significant damages (Leckebusch et al., 2007; Pinto et al., 2010).

During recent decades, Europe has faced highly impactful windstorms that have caused considerable human and economic
impacts, ranging from human fatalities to damage to infrastructures, agriculture and forest sectors, with estimated average



annual losses for the EU and UK amounting to 5 €billion/year (2015 values). The highest absolute losses have been in Germany (850 €million/year), France (680 €million/year), Italy (540 €million/year) and the UK (530 €million/year) (Spinoni et al., 2020).

Windstorms associated with ETCs present considerable inter-annual variability, hampering trend detection (Dawkins et al., 2016). In this context, relevant uncertainty affects the response of the North Atlantic storm track to a warmer atmosphere (Shaw et al., 2016; Woollings et al., 2012). If, on the one hand, a certain degree of confidence characterizes the response of ETC to the increasing atmospheric moisture content, on the other hand the weakening of the lower-tropospheric meridional temperature gradient and the enhanced tropical upper tropospheric warming construct a general picture where non-linear and possibly compensating factors can make future changes of ETC climate more complicated to evaluate, understand and predict

(Woollings et al., 2023). Nevertheless, North Atlantic ETC trend evaluation and understanding, as well as their consequences, crucially depend on methodological analysis choices regarding datasets (e.g., observations, reanalysis, proxies, model simulations and analysis period) and approaches to examine storm features (i.e., Eulerian vs. Lagrangian). The latest IPCC report highlights extreme wind projections characterized by large uncertainties (Ranasinghe et al., 2021; Seneviratne et al., 2023). Despite this, there is moderate confidence in recent projections which report an increase in frequency and intensity of

storms in Northern and Central Europe, combined with a decrease in their frequency over Southern Europe (Catto et al., 2019; Little et al., 2023; Priestley and Catto, 2022).

Concerning what climate change implies regarding storm-related losses, a few studies suggest an increase in Central and Northern Europe (Donat et al., 2011; Leckebusch et al., 2007; Little et al., 2023). There are, however, noteworthy challenges in translating changes in ETC features in losses, which result from multiple and intertwined risk factors, i.e. hazard, exposure

and vulnerability components (Flynn et al., 2024). In addition, a standardized and shared framework for describing scenarios, assumptions and results is still missing (Ranson et al., 2014).

Cataloguing ETC properties like intensity, location and frequency is crucial for our understanding of the factors that most influence their development, as well as for evaluating and improving the predictions from weather and climate models. ETC data are scarce and generally limited to well-known harmful events. In this regard, we consider as reference product the XWS

(eXtreme WindStorms) catalogue, consisting of storm tracks and model-generated wind-gust footprints for 51 of the most extreme winter windstorms to hit Europe in the period 1979–2012 (Roberts et al., 2014).

This work presents a novel dataset of ETC windstorm tracks and associated wind-gust footprints based on the entire time series available (1940-present) from the ERA5 reanalysis (Hersbach et al., 2020). The objective of this innovation is to promote a knowledge-based assessment of the nature and temporal evolution of windstorms associated with ETCs. The principal aim is

to assist users of the insurance sector, reinsurers and insurance industry service providers, in response to their requirements for a catalogue of historic windstorm events over Europe. This sector needs to be able to characterize the temporal and geographical distribution of potentially destructive windstorm events over the European territory. The data contained in such hazard catalogues can also support sectors such as forestry, agriculture, energy, transport, civil engineering and government. Businesses in those sectors can use the data for operational planning and mitigating windstorm-related risks.



The dataset has been produced, within the Copernicus Climate Change Service (C3S) "Enhanced operational Windstorm Service" (EWS), by the European Centre for Medium-Range Weather Forecasts (ECMWF)-implemented C3S2_413 Contract and represents a continuation, a temporal extension and an enhancement of the previously operational C3S Windstorm Service. C3S is one of the six services of Copernicus, Earth observation component of the European Union's Space programme, and it is operated by the ECMWF on behalf of the European Commission with funding from the European

Union. Copernicus provides open and free access to quality-controlled climate data and tools for use by governments, public authorities, and private entities around the world. EWS is based on two datasets: (i) windstorm tracks and (ii) the associated footprints. These are defined as follows:

- Windstorm tracks. The previous service data content is now extended to the detection and tracking of pan-European potentially harmful windstorms, associated with ETCs, employing the previously used TRACK algorithm (Hodges 1995, 1999,

Hoskins and Hodges 2002) and a second tracking algorithm, TempestExtremes (TE, Ullrich et al., 2021) allowing, in this way, by comparison, to estimate the uncertainty of the results.

- Windstorm footprints. Following the approach of the XWS catalogue (Roberts et al., 2014), a windstorm footprint is defined as the maximum 10m wind-gust at each grid point of the domain over a 72-hour time window. The time window is centred on the time that the tracking algorithm identified as having the maximum 925 hPa wind speed over land, within a 3-degree radius

of the track centre. Footprint datasets are produced considering original-resolution ERA5 variables and statistically downscaled ERA5 variables with a destination grid at 1 km resolution.

In addition to this, in the new EWS, the analysis, previously limited to the period 1979-2021, is now extended to the entire ERA5 dataset, 1940-2024, and is kept constantly updated in time, on a bi-monthly basis, by adding new ERA5 reanalyses as soon as they become available, in a "quasi-operational" fashion. EWS also includes two Windstorm Applications, designed to

guarantee an easy and customizable exploration of the datasets. Such Windstorm Applications offer high-level data selection and visualization capabilities, enabling a single event or spatially/temporally aggregated assessment of the datasets. This enables effective access to the essential windstorm features, without requiring the users to download the full EWS datasets.

The manuscript develops as follows: in section 2, windstorm datasets are defined and presented. An evaluation of windstorm tracks and footprints is presented in section 3, where the problem of reproducibility of tracks, relevant to industry users, is also

treated, while a trends analysis is presented in section 4. In section 5, the windstorm summary indicators and the EWS applications are described and in section 6 some conclusions are drawn.

## 2 The EWS datasets

In this section, we describe the methodological steps taken in the production of the two datasets contained in the EWS.



### 2.1 ETC tracks

The first EWS dataset consists of ETC windstorm tracks from 1940 to 2023, detected using two different algorithms. TRACK and TE algorithms leverage two different atmospheric tracking variables: 850 hPa relative vorticity and mean sea level pressure (MSLP), respectively. These are two of the often used Lagrangian approaches to assess storm track spatial and temporal evolution (Hoskins and Hodges, 2002; Neu et al., 2013; Walker et al., 2020) and reflect different aspects of synoptic dynamics, influencing tracks' detection. Even though tracks climatological features defined through vorticity- and MSLP-based

approaches are generally similar, vorticity is found to better capture the small-spatial-scale end of the synoptic scale range. This translates into more cyclonic systems identified in areas, e.g., the Mediterranean basin, than in the MSLP which focuses on synoptic large-scale end (Hodges et al., 2003).

Comparing two tracking algorithms allows one to glimpse basic uncertainty about track climatological statistics and also to explore differences in the underlying individual events detection and representation, as a function of the tracking algorithm

selected. Nevertheless, the two algorithms also differ in the preprocessing steps contributing to the overall track features variability. These latter are mainly represented by the spectral truncation and specific planetary wave numbers filtering required by the vorticity-based tracking, described in what follows.

### *The TRACK algorithm*

In continuity with the previous version of the Copernicus Windstorm Service (Copernicus Climate Change Service, Climate Data Store, 2022), the first tracking algorithm considered in the EWS is the TRACK tracking algorithm (Hodges 1995, 1999, Hoskins and Hodges 2002). TRACK is implemented in the same configuration as in the previous service, but extending the period considered to encompass the whole ERA5 period (1940-2023). TRACK uses a 3-hourly frequency of 850 hPa relative vorticity as the variable to detect and track the low-pressure system. This input variable is provided in ERA5 on the native IFS

Gaussian grid (N320) of the assimilating model. The following steps are then performed:

- The 850 hPa relative vorticity fields are spectrally filtered to T42 resolution (corresponding to about 480 km) to eliminate features related to small-scale spatial variance. Post detection and tracking, the full resolution fields (vorticity, winds, MSLP) are added back to the tracks.

- As part of the initial detection prior to tracking relative vorticity maxima with a value larger than $1\times10^{-5}s^{-1}$ are retained.

This is performed on a polar stereographic projection and the off-grid locations are determined using B-spline interpolation and steepest ascent maximisation to produce more accurate locations.

- The adaptive algorithm described in Hodges (1999) is then applied, using settings suitable for ETCs and 3 hourly data, to merge vorticity features into cyclone tracks.

- Tracks that last less than 1 day or travel less than 1000 km are discarded.

- The mean sea level pressure and 10m wind speed fields are added by searching for the nearest pressure minimum with a 5° radius from the vorticity centre using B-spline interpolation and steepest descent minimisation.



*The TEMPEST_EXTREMES (TE) algorithm*

The TempestExtremes algorithm (TE, Ullrich et al., 2021) is the second tracking algorithm used for ETC tracking. Differently
from TRACK, the TE tracking algorithm is based on 6-hourly Mean Sea Level Pressure (MSLP). TE first applies a cyclonic
storm selection criterion such that the minimum cyclone pressure must be enclosed by a closed contour of 200 Pa within 6.0°
of the cyclone centre. The location of the minimum pressure defines the centre of the cyclone. Candidates within 6.0° of one
another are merged, with the lowest pressure having precedence. Outputs from this first pass are then concatenated into a single
candidate list, tracking these features in time. In our configuration, the track nodes stitching function requires:
- Storms persisting for at least 60 hours.
- A maximum gap (time between sequential detections satisfying the nodes detecting criteria) of at most 18 hours.
- Finally, the TE algorithm imposes that the ETC moves at least 12° from the start to the end of the trajectory to eliminate
stationary features (e.g., the Icelandic Low) and spurious shallow lows generated in the lee of high topography.

With the aim of maintaining a manageable number of events and focusing primarily on those most likely relevant to the
insurance sector, raw windstorm track candidates produced by the two algorithms are filtered according to spatial and temporal
criteria. Firstly, within a specific event, only time steps having MSLP <=990 hPa are kept, as generally capable of generating
hazardous windspeeds. Then, events are filtered out if they show at least one of the following features:
Tracks with less than 5 points; tracks with more than 3 points north of 65 °N; tracks starting eastward of 5°W; tracks ending
westward of 5°W. Nevertheless, all original tracks produced by the two algorithms (without footprints) will be made available
to the users.
The catalogue entry with the list of tracks which have passed the filtering criteria consists of text-like [.csv] files. Each track
is identified by: ID; longitude and latitude; mean sea level pressure at each step; date; wind gust value at 10m height; land-
sea-mask reporting the percentage of land area of the track grid points. All the above-mentioned variables are provided at full
resolution.
Figure 1 shows a comparison between the tracks produced by the two algorithms, for the entire year (irrespective of season)
and in terms of the corresponding MSLP computed for each segment constituting a track. All the tracks detected by a single
algorithm are shown in panels (a) and those detected only by a single algorithm are shown in panels (b). Left panels TRACK,
right panels TE respectively. An event is (arbitrarily) labelled as detected by both algorithms when at least half of the track
points are detected by both, namely it shows a temporal and spatial discrepancy not higher than one day and three degrees of
latitude and longitude respectively.



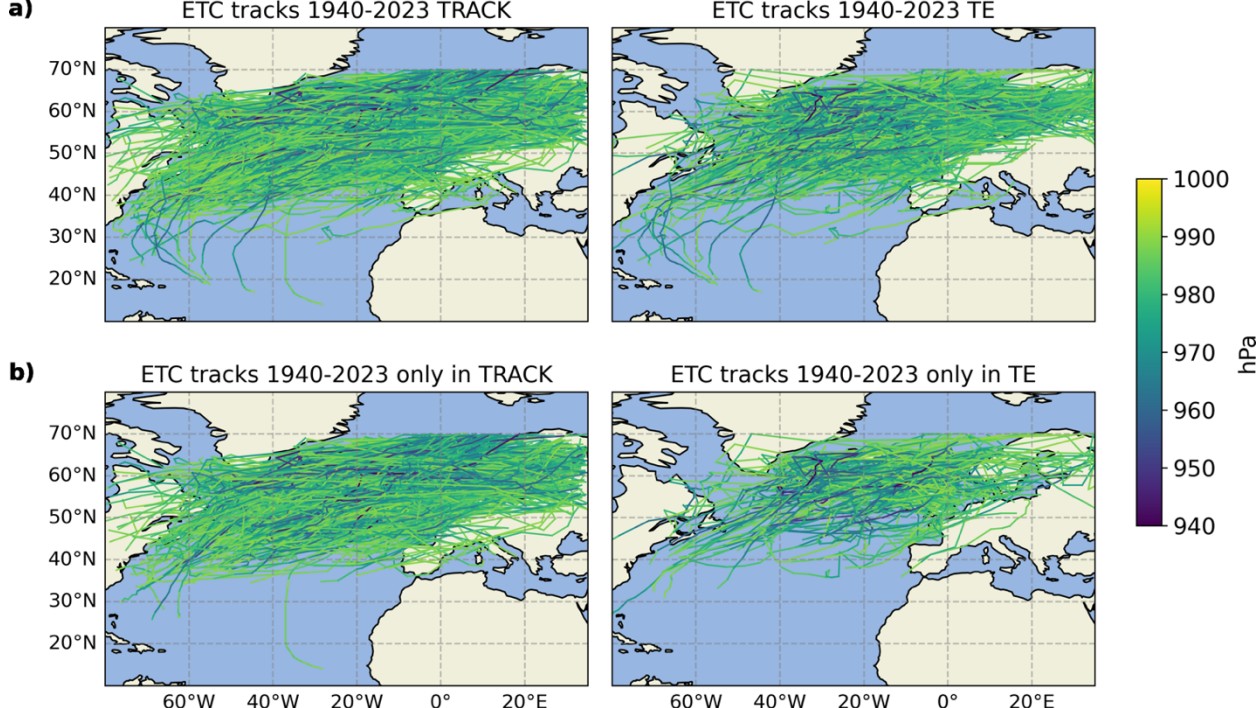

**Figure 1. Comparison of ETC tracks with the associated MSLP produced by the two different tracking algorithms: TRACK (left column) and TempestExtremes (TE, right column), from 1940 to 2023 based on ERA5. The bottom panels show tracks exclusively detected by TRACK and by TE.**

The TRACK algorithm tends to produce a considerably larger number of events (1845 events against 847 events detected by TE). Noteworthy is the fact that the considerably fewer events detected by TE are not all necessarily included within the TRACK events set, as shown in Figure 1b, where events that are detected by both algorithms are removed.

Figure 2 shows bi-dimensional density functions of two climatological features of the tracks, detected by two metrics, distance travelled and persistence in time, characterizing individual events during the whole climatological period. All the events and exclusive events detected by one tracking algorithm are shown in subpanel (a) and (b) respectively. The two algorithms show tracks with comparable spatial extension but with TE showing a generally longer temporal persistence of the windstorm tracks. Moreover, TRACK is quite consistent with itself, while TE shows a strong difference between all versus TE-only. This difference points out that the influence of a different tracking variable is not limited to generating a different number of tracks but also to selecting events with different characteristics. In this regard, several studies highlight how the tracking of different fields can substantially alter the number of detected cyclones, even when considering the same dataset (Feser et al., 2015; Hoskins and Hodges, 2002; Neu et al., 2013).






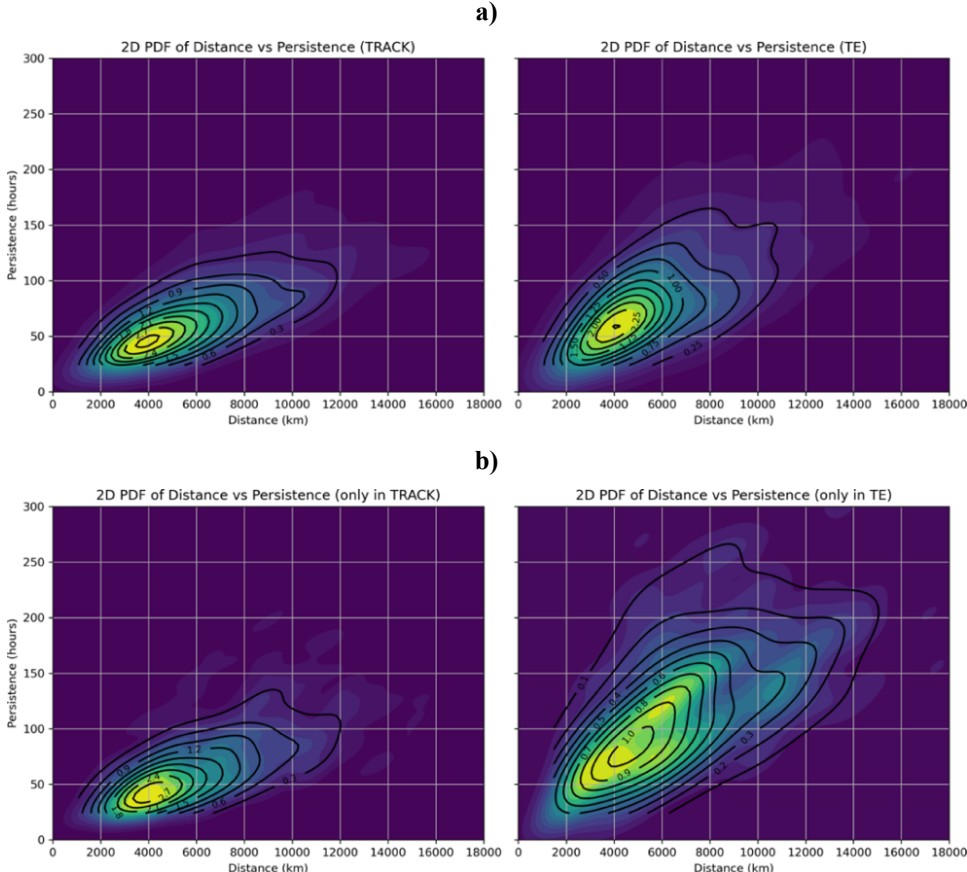

**Figure 2. Comparison of bi-dimensional density built on distance and persistence of tracks detected by the two algorithms. Top panels (a): all tracks from TRACK (left panel) and from TempestExtremes (TE, right panel), from 1940 to 2023 based on ERA5. Bottom panels (b): tracks exclusively detected by TRACK (left panel) and only by TE (right panel).**

Figure 3 shows basic windstorm features for the cold (a) and warm (b) semesters. These are represented by mean MSLP and the spatial density of windstorm tracks crossing the domain divided into cells of 4ºx4º, during the period 1940-2023. The number of windstorm tracks crossing each cell is counted once, even when it persists or crosses the same cell in more than one time step. During the cold semester, tracks generated by TE tend to show fewer events, albeit characterised by lower mean

MSLP, with the minima shifted westward compared to TRACK. A similar difference, even though to a lesser extent, can be observed in the warm semester. Besides mean MSLP, a relevant difference between the two algorithms is the non-negligible higher number of ETC track passages detected by TRACK compared with TE, although producing similar overall spatial patterns in both semesters.






**a)**

**b)**

**Figure 3. ERA5 ETC tracks MSLP and spatial distribution of number of tracks in each cell of 4° in the period 1940–2023 for the cold (a) and warm (b) semesters. Windstorm tracks produced by the TRACK (TE) algorithm are shown on the left (right) panels.**

## 2.2 Windstorm footprints

For all diagnosed tracks, an associated footprint is derived according to the approach proposed in (Roberts et al., 2014). A storm footprint is defined as the maximum 10m-height wind gust at each grid point in the domain over a 72-hour time window. The time window is centred on the time that the tracking algorithm identified as having the maximum 925 hPa wind speed over land, within a 3-degree radius of the track centre (Figure 4). Footprints are thought to be a quantity of particular relevance to the insurance sector because they convey information about storm intensity, and therefore potential damages. Representative



events in Figure 4 result remarkably similar in the two algorithms despite the MSLP, variable used to track the event, representing a prognostic variable in TE (MSLP minimum in correspondence of grid points) and in TRACK diagnostic (minimum sought using the B-spline and minimization method to find the off-grid location). This similarity translates into

small uncertainty characterising these two event tracks and footprints.

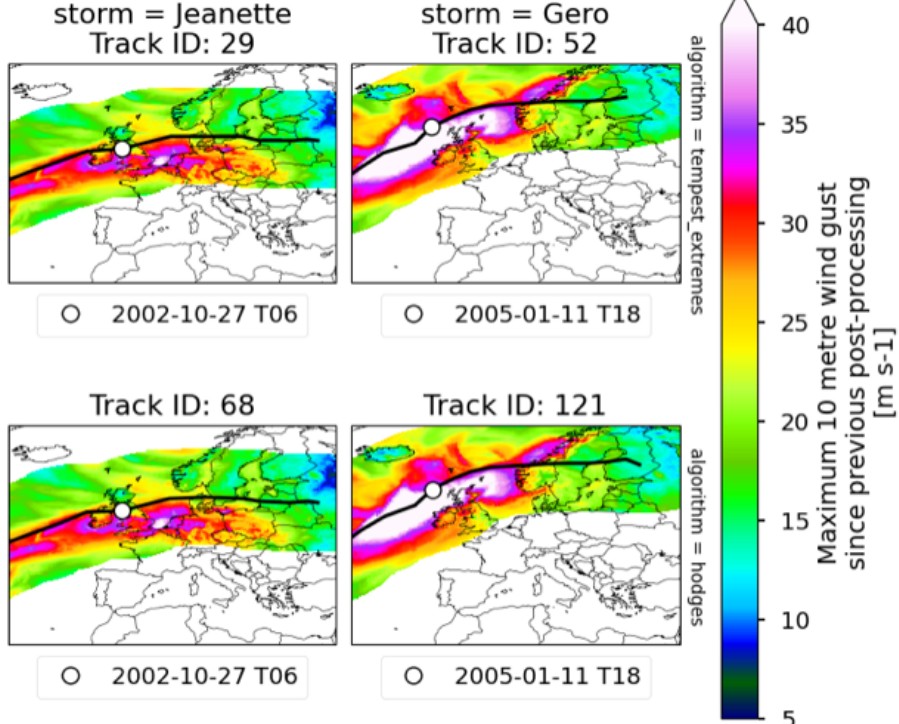

**Figure 4. Two examples of footprints for two representative windstorms (Jeanette and Gero) tracked with the two algorithms, TE and TRACK, are shown in the upper and bottom panels respectively. Given the different number of detected tracks, the same event corresponds to a different ID in the two tracking algorithms.**

Similarly to what is shown for the tracks in Figure 2, Figure 5 shows a bi-dimensional distribution of two footprints metrics: mean intensity (i.e., wind gust) and mean spatial extension considering all the events detected by the two algorithms. Here, it is noteworthy similar densities in the main core of the inner contours, but with TRACK showing more density in the top lobe (largest extension). This analysis considers only original resolution and decontaminated footprints (see inherent definitions of

decontamination and downscaling in the following sections) and highlights similar climatological features of the footprints associated to the tracks identified by the two algorithms.



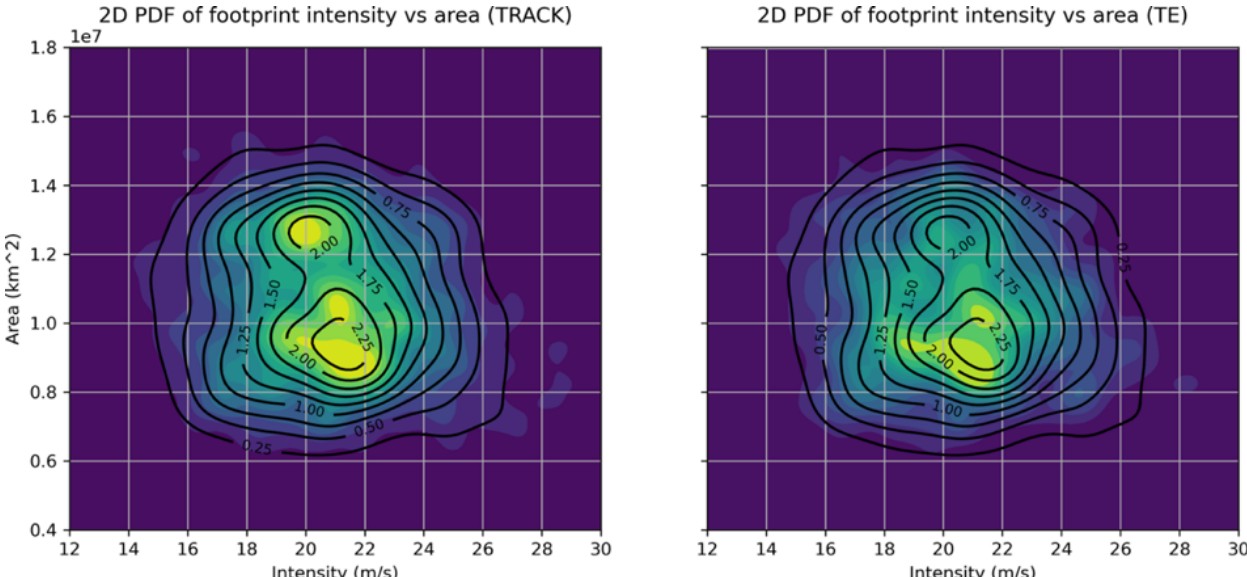

**Figure 5. 2D PDF of windstorm footprint intensity vs. affected area means considering the whole ERA5 period.**

## 2.3 Footprints statistical downscaling

Wind gust represents a climate variable resulting from complex interactions between dynamical flow and local-scale

physiography (topographical, coastline, etc.) features. ERA5 reanalysis has a resolution of approximately 30km, still relatively coarse for reproducing fine-scale processes. This, in general, translates into an underestimation of wind gust maxima. Surface level wind gusts can also be estimated from the shear of the wind speed between the two lowermost ERA5 levels, in short wgSLh, which is an abbreviation for wind gusts estimated from shear and the logarithmic of the heights. Here we make use of a multiple linear regression-based statistical downscaling approach (van den Brink, 2020) as in the existing windstorm service

dataset, and in contrast to the dynamical downscaling used for generating XWS footprints dataset. Statistical downscaling combines these two sources of wind gust information, as well as surface elevation, to derive high-resolution gridded estimates of footprint wind gusts:

$$WindGust = 10.3 + 0.0112 * ERA5^2 + 0.0124 * wgSLh^2 + 0.00355 * ELEV$$

$$wgSLh = u_{10} + \alpha \frac{u_{100} - u_{10}}{log(100/10)}$$

Where ERA5 denotes predictor 1 (10m wind gust); wgSLh denotes predictor 2 (wind gust estimated from wind shear between two ERA5 levels; ELEV denotes predictor 3 (observed elevation derived from a 1 km resolution terrain height map); $\alpha$ is the median of the normalized gust distribution, with a value of 3.25 (van den Brink, 2020).



## 2.4 Footprint decontamination

European storms can cluster in time and footprint wind gusts could likely be derived from two or more events. Following
Roberts et al. (2014), to minimize this "contamination", instead of taking the maximum gust over the whole domain, only gusts
inside a 1000km radius of the track position at that time are assumed to be part of the event (Figure 6).

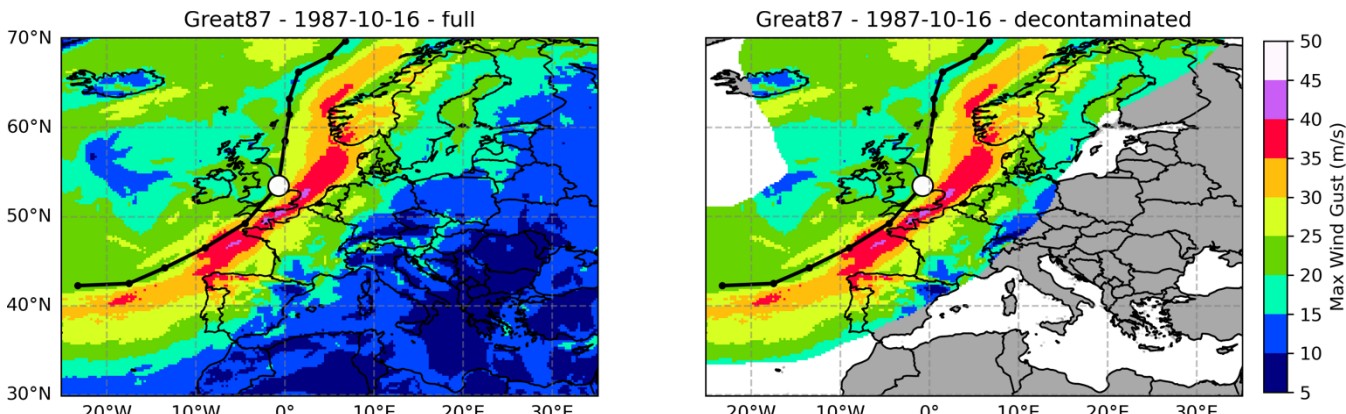

**Figure 6. An example of a windstorm (the Great87) footprint, original and decontaminated, left and right panels respectively.**

## 3 Evaluation of the reproducibility of the windstorm tracks and footprints

In this section, we provide an introductory evaluation of the reproducibility of the windstorm tracks and footprints datasets.
For what concerns the windstorm tracks, the reference product is the XWS catalogue (Robert et al., 2014). The reference
product used to evaluate the windstorm footprints is HadlSD, a quality-controlled, sub-daily station dataset with data from
1931 to June 2024 (Dunn et al. 2016). HadlSD provides 10m-height wind gusts allowing us to evaluate the plausibility of the
ERA5 footprint wind gusts.

### 3.1 Evaluation of the Windstorm Tracks

Here, we evaluate the reproducibility of a set of 51 historical windstorms included in the XWS catalogue. The reproducibility
consists of identifying shared events between the two tracking algorithms and the reference XWS tracks. The same inter-track
matching parameters mentioned in section 2.1 are here adopted defining (i.e., at least half points with a temporal and spatial
discrepancy not higher than one day and three degrees respectively). XWS tracks are produced considering ERA-Interim
reanalysis data (Dee et al. 2011) covering 34 extended winters (October to March) from 1979 to 2013. Noteworthy is the fact
that the tracking algorithm employed in the XWS dataset is TRACK with the same configuration as used here and using 3
hourly data. This entails a somewhat non-neutral approach, since the TRACK algorithm is also one of the two tracking
algorithms we use, albeit applied to the ERA5 dataset, to produce the EWS tracks, in continuity with the previous Windstorm
service. Notwithstanding this, differences between the XWS and EWS tracks derived with TRACK can arise from differences



between the two different reanalysis releases, including a different resolution (~75 km for ERA-Interim vs. ~30 km for ERA5) and from the different versions of the TRACK algorithm used for XWS and EWS. In Figure 7 we illustrate three windstorms

from the XWS catalogue (Daria, Fanny and Gero) as reproduced by the two algorithms. We show both filtered (see filtering strategy in section 2.1) and raw tracks to provide a basic idea of the impact of the filtering strategy. However, the latter can vary substantially depending on the reference event. For this reason, raw traces (without filtering) will also be provided within the EWS.

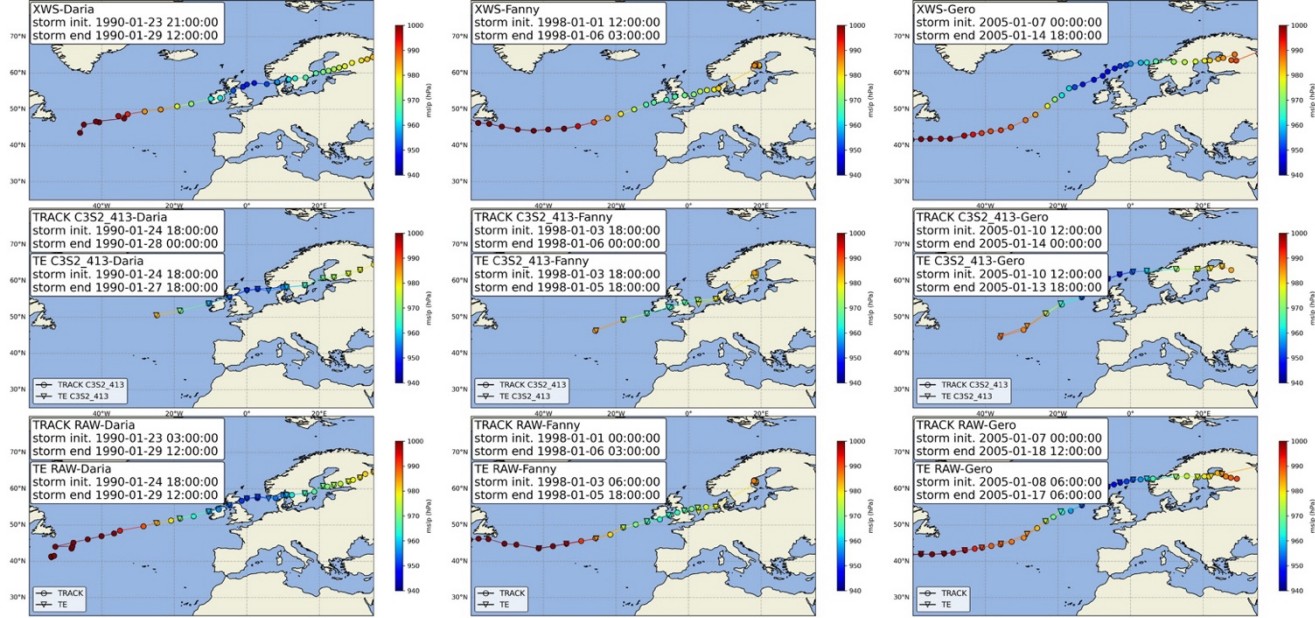

**Figure 7. Windstorm tracks for three named storms from the XWS reference dataset (upper panels) and from the two tracking algorithms (TRACK and TE) with and without filtering in the central and bottom panels respectively.**

The upper panels show XWS tracks with the corresponding name initial and final time step date. The same is shown in the

central panels for the track produced by TRACK and TE. Finally, in the bottom panels, the same tracks but with no filtering applied. The colour of the track indicates the value of the MSLP in the specific timestep, for the three different windstorm tracks. Generally (beyond the three tracks shown here), the majority of the XWS windstorms are reproduced by tracking algorithms, 40 cases out of 51. In TRACK, missing events are those not satisfying filtering parameters (sec. 2.1) since found in the raw tracks. In TE, among those not presented in the EWS dataset are present a few events not represented even in the

raw tracks (Table SM1). As already mentioned, a much higher similarity between XWS and TRACK than TE (40 and 18 respectively) is expected because of the use of the same tracking algorithm. From a purely user perspective, we advocate combining both tracking algorithms to maximise the characterisation of windstorm events (e.g., Moemken et al, 2024).



### 3.3 Evaluation of the Footprints

Footprints have been produced for all windstorm tracks generated by the two tracking algorithms and evaluated against the
HadISD observational dataset available during the time segments in which windstorms occurred. The procedure applied to the
HadISD dataset to produce a reference footprint begins by selecting, among the 7677 available stations of the dataset, those
that provide the 6-hourly 10m height wind gust variable, over the pan-European domain, during the time steps identified by
the presence of a selected windstorm event. As done with ERA5, but for each observational site, instead of each grid point, the
maximum of the wind gust during the 72-hour time window is derived. Here, an arbitrary threshold of at least 10 valid values
(out of the 12 6-hourly time steps within the 72-hour time window) is applied per reference site and event, from which to
derive the maximum wind gust. In the evaluation, only two of the four types of footprints defined are considered, namely the
ERA5-original-resolution-decontaminated   and   the   statistically-downscaled-decontaminated   footprints.   For   the   sake   of
conciseness, only a small subset of original-resolution and downscaled decontaminated footprints, corresponding to events
included in the XWS dataset, are shown here. The windstorm tracks shown in these examples are defined with the TRACK
algorithm. Figure 8 shows a general overview of the evaluation process, with footprints associated with tracks from the
TRACK algorithm from three named events (XWS Catalogue). Results from the original resolution and downscaled ERA5
are reported. In the right column, all the HadISD reference sites with available wind gusts during the 72-hour time window
used to derive the footprint are shown.

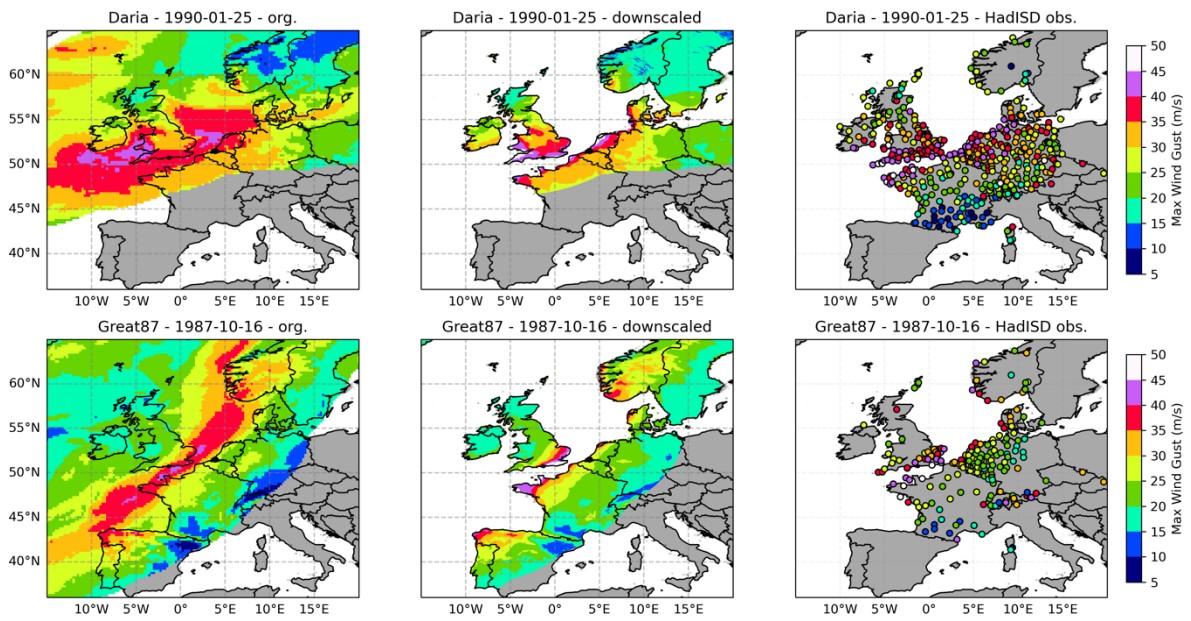



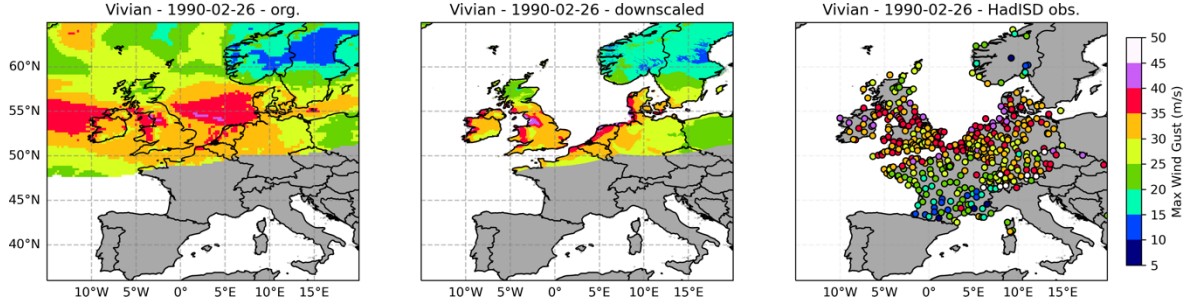

**Figure 8. Comparison, for three named events of the ERA5-based footprints with the footprints computed from the point-scale observational sites of the HadISD dataset. The left column shows original resolution-decontaminated footprints, the central column shows downscaled-decontaminated footprints (only over land by construction), and the right column maps show the geographical locations of the reference observation sites and the strength of the observed wind gusts maximum. The colour code is the same for all the three panels.**

From Figure 8, ERA5 and observations tend to agree reasonably well on the footprint spatial patterns, especially over the
British Isles, e.g. along the southern coast, else windward vs leeward of the Pennines. However, for a few representative storms, an apparent overestimation of the wind gust is produced by ERA5 over the Netherlands, considering both original and downscaled footprints. In addition to a general and expected orography-driven refinement of the wind gust spatial patterns, downscaled ERA5 wind gusts show generally, though not systematically, higher values, which, in general, compare better with observations. This appears to happen regardless of the original wind gust value from the original ERA5 data. This effect
can be observed over the Scandinavian peninsula, and in the area affected by the strongest wind gust during the Daria storm between the UK and the Netherlands. Other examples of stronger wind gusts in the downscaled ERA5 are represented by the Great87 storm over Britain and southeastern England and the u19840113 storm (not shown) over central UK and Denmark.

In a second analysis (Figure 9), in order to concentrate on more intense windstorms, likely to cause more damage, we select only wind gusts above 15 m/s and associated biases, focusing on a direct comparison of the original resolution and the
statistically downscaled ERA5 products with HadISD reference site observations. Here, we extract the wind gust values from the nearest ERA5 grid point to the observational sites. Figure 9 shows the results for the same three named events as for Figure 8, with the original and downscaled ERA5 nearest grid point wind gust (left and central columns respectively) compared to the observed wind gust of the reference sites (right column). Despite the intrinsic limitation of comparing nearest grid point values with sparse observation point reference sites, it can be observed that the downscaling procedure produces generally
improved data. The wind gusts associated with the Daria, Great87, Vivian and most other events (not shown) are improved in the downscaled ERA5 over a large portion of the domain affected, by reducing the underestimation characterizing the original resolution ERA5. The English Channel and North Sea coasts are the domain portions more frequently and more largely benefiting from downscaling in terms of wind gust reproduction. However, downscaling improvement is not systematic. For instance, wind gust overestimations over a few points on the Netherlands coast during the Daria storm can be observed.
The smaller number of grid points selected in the original resolution vs. the downscaled dataset (left and central column in Figure 9 respectively) is caused by the nearest-neighbour selection of grid points. Sea grid points are preliminarily removed



from the original resolution footprints for consistency with the downscaled footprints (with no sea points by construction). Given the coarser resolution, original resolution footprints have more cases where nearest neighbours fall in sea grid points and then, are not considered.

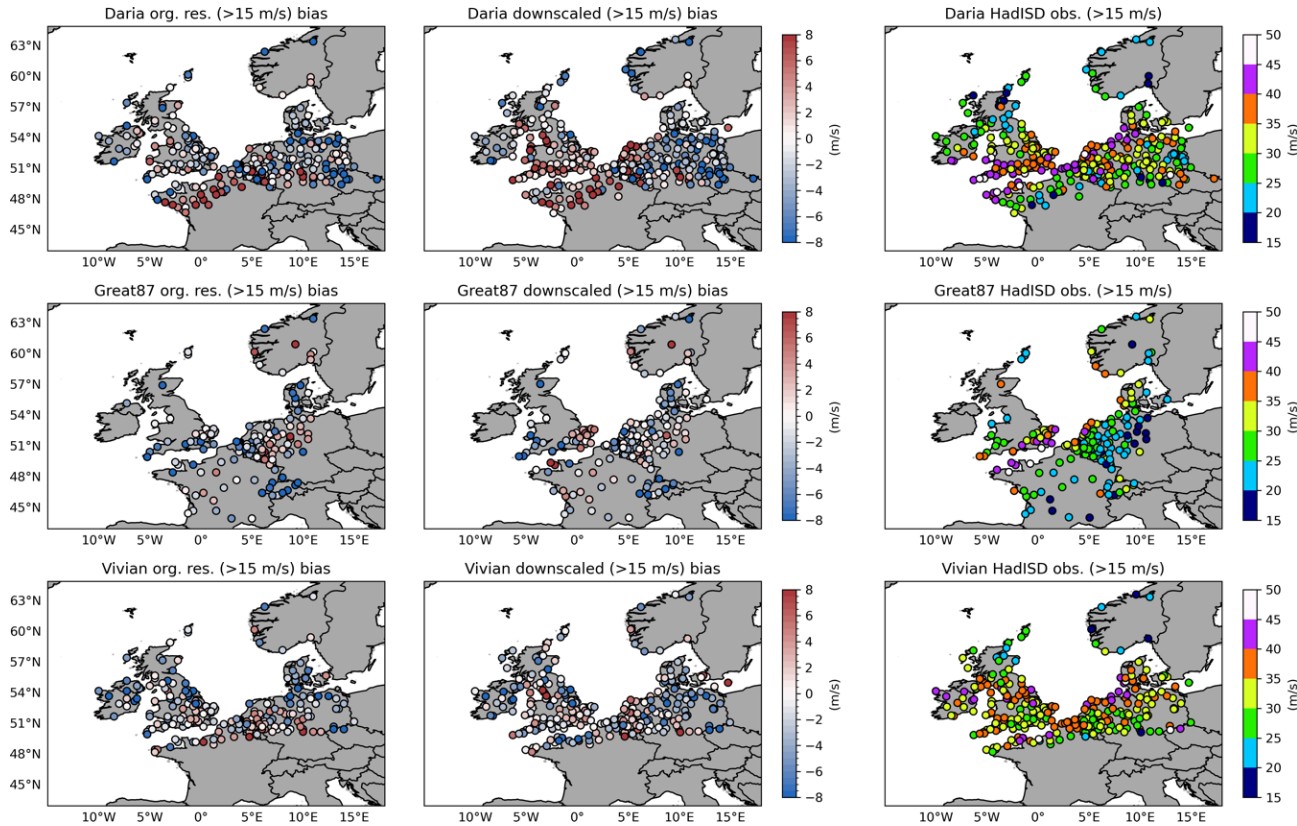

**Figure 9. Footprint wind gusts (>15 m/s) bias from the original resolution (left column) and downscaled (central column) ERA5 grid points nearest to the reference sites (right column) for three representative events.**

Evidence pointing toward a moderate improvement in the downscaled footprints can be also observed through the PDF-based analysis shown in Figure 10. Here, PDFs are built considering the same grid points from the original ERA5 resolution, downscaled ERA5 and reference sites previously shown in Figure 9. The PDF histograms have a fixed bin width of 2.5 m/s and are combined with an estimate of the PDF based on the Kernel Density Function, shown with solid lines. Here, the general, though not systematic, improvement of the downscaled ERA5 wind gusts is confirmed, especially regarding the right tail of

the distribution, representing the most extreme values.



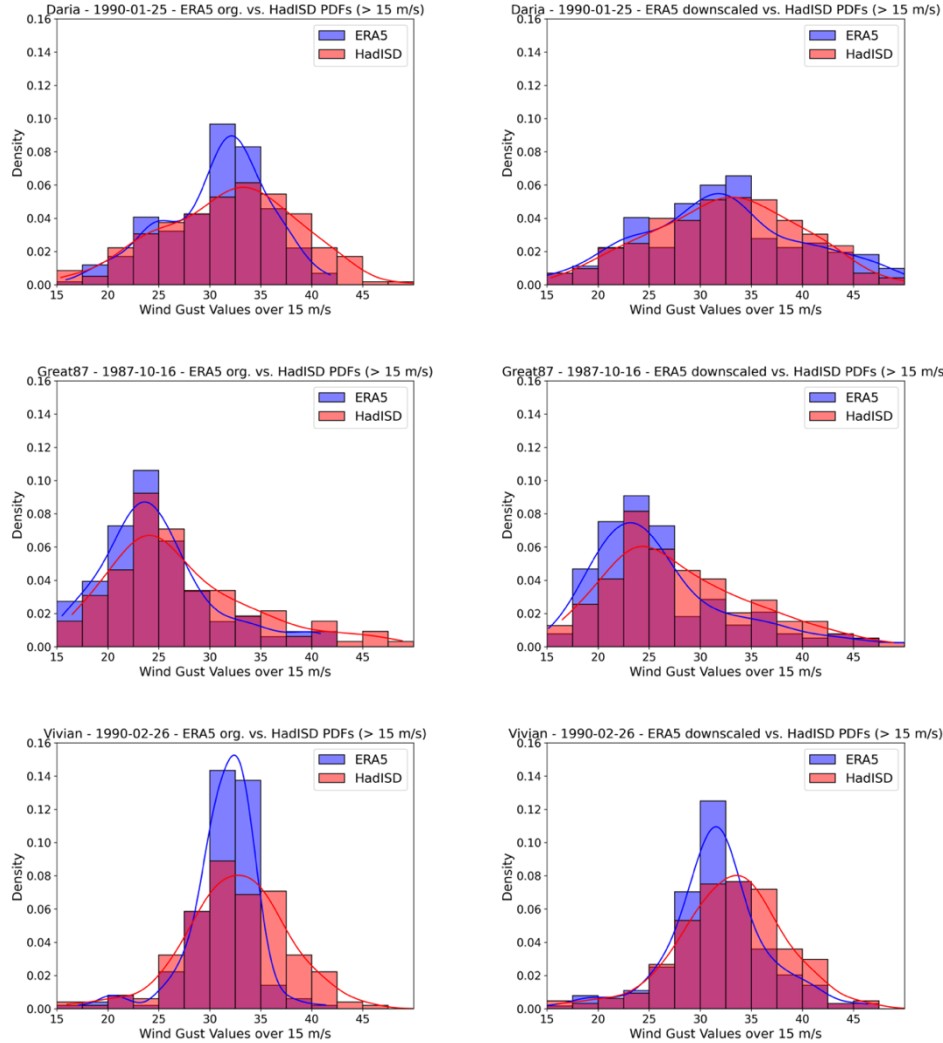

**Figure 10. PDFs built on original (left column) and downscaled (right column) ERA5 wind gust compared to the observations (HadISD). The underlying grid points and reference sites underlying the PDFs correspond to those shown in Figure 9.**

## 4 Windstorm track and footprint trends

In this section, we examine trends in time characterising windstorm tracks and associated footprints, considering all the windstorms detected and tracked with the two algorithms during the whole period available from 1940 to 2023. In this context, a non-negligeable source of uncertainty when using reanalyses over such extended periods arises from changes in the observational data assimilated into the reanalysis system, here not considered. As for the analysis shown in Figure 3, the whole domain is divided into cells of 4°x4°, binning all the track points (time steps) in the closest grid cell. After that, we perform an annual temporal aggregation for the cold and warm semesters: (i) averaging the mean sea level pressure of the track points and



(ii) counting unique track points (i.e., estimating spatial density), defining in this way two grid-point-specific time series (with 84-time steps). Figure 11 reports the results of windstorm track spatial density (number of tracks/year) trends derived through

a Mann-Kendall test.

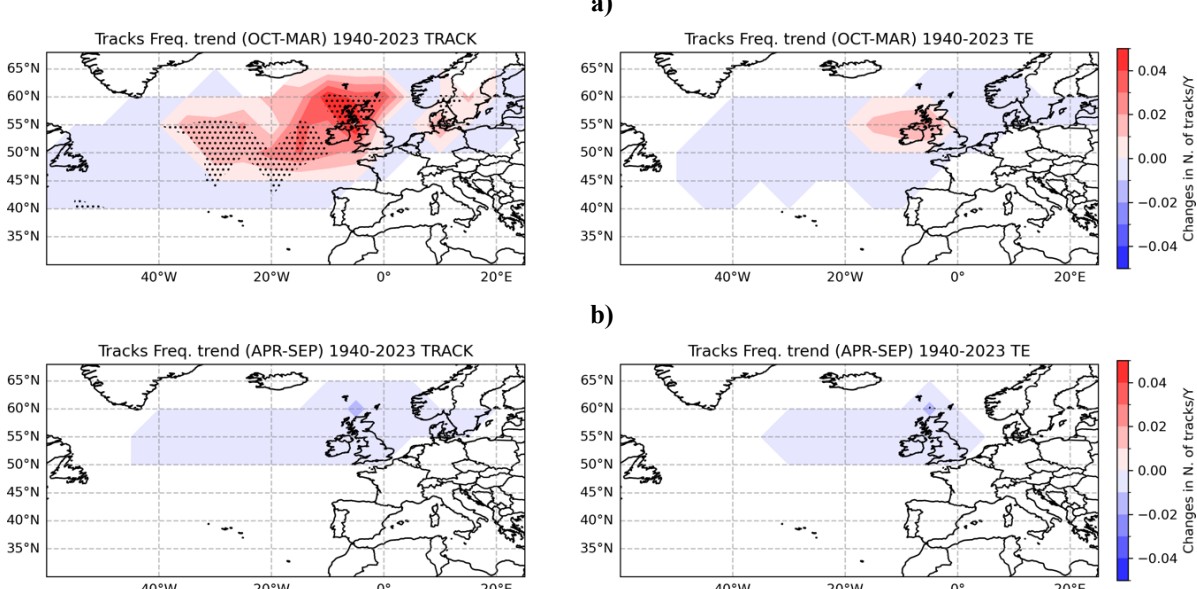

**Figure 11. Trends of ETC tracks number for the cold (a) and warm (b) semesters. Grid points with significant trends (Mann-Kendall test p-value lower than 0.05) are stippled. Results from TRACK and TE algorithms are shown in the left and right panels respectively.**

Figure 11 shows the results of the trend evaluation as a function of the semester and of the tracking algorithm considered. First, both algorithms identify increasing trends over parts of the North-Eastern Atlantic only during the cold semester consistently to the review study by Feser et al. (2015). At the same time, TRACK shows an enhanced trend compared to TE in terms of spatial extension, trend magnitude and statistical significance. The results of the same analysis on tracks mean MSLP did not

yield any noticeable trend and is therefore not shown. Regarding footprints, we similarly aggregate wind gust values for each year, grid point by grid point, considering original and downscaled ERA5. For each year, we derive annual means and means for the cold (Oct-Mar) and warm (Apr-Sep) semesters. As for the tracks, a trend via the Mann-Kendall test is derived for each grid point time series. Figure 12 shows cold and warm semesters, which are characterized by rather different trends. On the one hand, the Scandinavian peninsula shows increasing mean footprint wind gust values in both semesters, although more

intense in the cold semester. Over the British Isles, instead, a trend of opposite sign is present in the two semesters, with a significant increase during the cold semester and a significant decrease during the warm semester. More generally, the cold semester shows a remarkable and widespread significant increase over Central and Northern Europe, whereas the warm semester shows less pronounced opposite trends, decreasing over the central British Isles and increasing over Sweden, both statistically significant. Concerning the impact produced by the statistical downscaling, the trends of the spatial patterns and



their statistical significance are preserved moving from the original to the downscaled ERA5 data, even though a general damping of the trend magnitude can be observed in the downscaled ERA5 footprints.

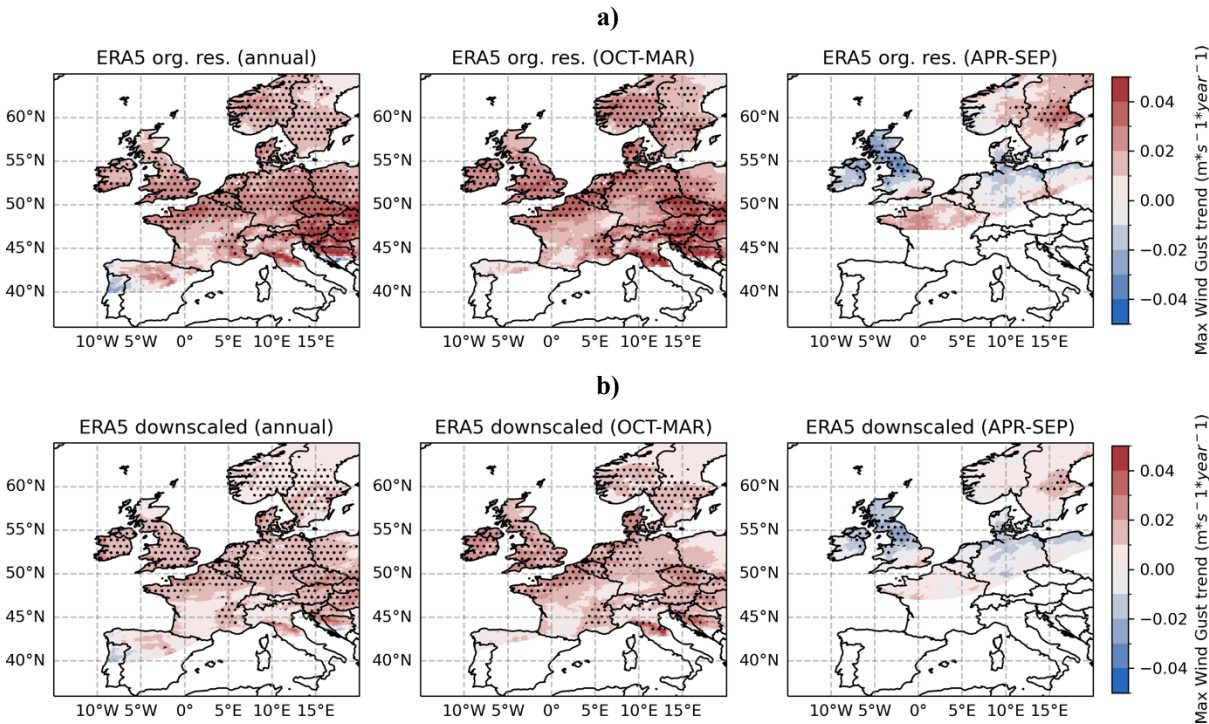

**Figure 12. Footprint wind gusts trend for original (a) and downscaled (b) ERA5. Annual mean trends are reported in the left sub-panels, cold and warm semesters in the central and right sub-panels respectively.**

## 5 Windstorm summary indicators and applications

Embracing the C3S objective of simplifying data discovery and access (Buontempo et al., 2022, Dee et al., 2024), EWS offers two windstorm applications, designed to guarantee a rapid and manageable exploration of the datasets. Windstorm applications
offer high-level data selection and visualization capabilities, enabling the assessment of a single event or spatially/temporally aggregated ensembles of events, up to entire datasets. This enables effective access to the essential windstorm features, without requiring the users to download entire EWS datasets. The first application serves as an exploration of EWS windstorm tracks and associated footprints, whereas the second provides several summary indicators. In the following sections, we describe the formulation of the EWS summary indicators describing, at the same time, the conceptualization and capabilities of the
applications.



## 5.1 Windstorm Summary Indicators (WSIs)

WSIs are computed yearly and are based on decontaminated footprints identified with both tracking algorithms. Three progressively higher wind gust thresholds (15, 20, and 25 m/s) are considered. Spatially, a progressively more refined subdivision of the European territory is available, following NUTS (Nomenclature of territorial units for statistics, https://ec.europa.eu/eurostat/web/nuts/overview). NUTS divides EU countries into 3 levels: NUTS 1: major socio-economic regions; NUTS 2: basic regions; NUTS 3: small regions.

WSIs are defined as follows:

- Yearly storm count. It counts the occurrence of storms exceeding the chosen threshold in the chosen region. A storm event is considered to affect a region if its decontaminated footprint intersects the region.

- Mean wind gust. It defines the average wind gust for storms exceeding a given threshold in a given region. For a given year, all the decontaminated footprints are filtered for each threshold and then averaged over time. The resulting mean wind gust speed is then spatially aggregated in the predefined NUTS. Areas of a region that are not affected by a storm are considered as "no values" and are ignored when performing the spatial average.

- Storm Severity Index (SSI, Dawkins et al., 2016). It aims at quantifying the severity of the storm affecting a region. SSI combines both the total area of a region affected by a storm in a year and the mean wind speed gust speed exceeding the threshold. The SSI can be summarised by the following formula:

$$SSI(threshold) = \cup A_{footprint \cap region} \left[ \overline{mean(windgust > threshold)} \right]^3$$

Where: $\cup A_{footprint \cap region}$ is the total area of the region affected by storms in a year. This area considers the union of the areas affected by every single footprint. $\overline{mean(windgust > threshold)}$ is the temporal mean of the wind gust speed exceeding the specific *threshold* spatially averaged over the selected region. The overline stands for the spatial average and is performed after the temporal average.

- Normalised Storm Severity Index (NSSI). Aims at quantifying the severity of the storm affecting a region with a dimensionless number. The NSSI is based on the SSI and introduces two factors allowing to normalise the SSI in terms of spatial extent and wind gust climatology statistics. The spatial normalisation consists of dividing the total area affected by storms by the area of the chosen NUTS region, so that the portion of the region affected by storms is accounted for, instead of the total area. The statistical normalisation consists of dividing the mean wind gust speed by the 98th percentile of the total wind gust probability distribution function (PDF, 1991 – 2020 period considered). This normalisation aims to modulate the SSI of single storms with the climatological severity of storms typical of a specific region (Little et al., 2023). It represents what historically, meaning climatologically, identifies as a severe wind gust for the specific country/region (according to the NUTS considered). The 98th percentile is different for different territories, reflecting the historically different exposure to severe wind gusts in the two countries (climate risk). This, in principle, should correspond to a





structural capability to cope with what the PDF defines as a severe wind gust. The NSSI is therefore derived using the following formula:

$$NSSI(threshold) = \frac{\cup A_{footprint \cap region}}{A_{region}} \left[ \frac{\overline{mean(windgust > threshold)}}{\overline{P_{98}}} \right]^3$$

Where: $A_{region}$ is the total area of the region (it is constant in time). $\overline{P_{98}}$ is the 98th percentile of the wind gust speed probability distribution considering all the decontaminated footprints over the period 1991 – 2020. The overline stands for the spatial average. In Figure 13 we show the NSSI time series from 1940 to 2023, spatially averaged over different European regions. Based on the footprint dataset, we can see an increasing trend for all four representative regions in agreement with the previous footprint trend analysis (Figure 12). However, we can also observe that only two out of the four regions show a statistically

significant trend (p-value < 0.05), that is Normandie (Northwestern France) and Southern Ireland.

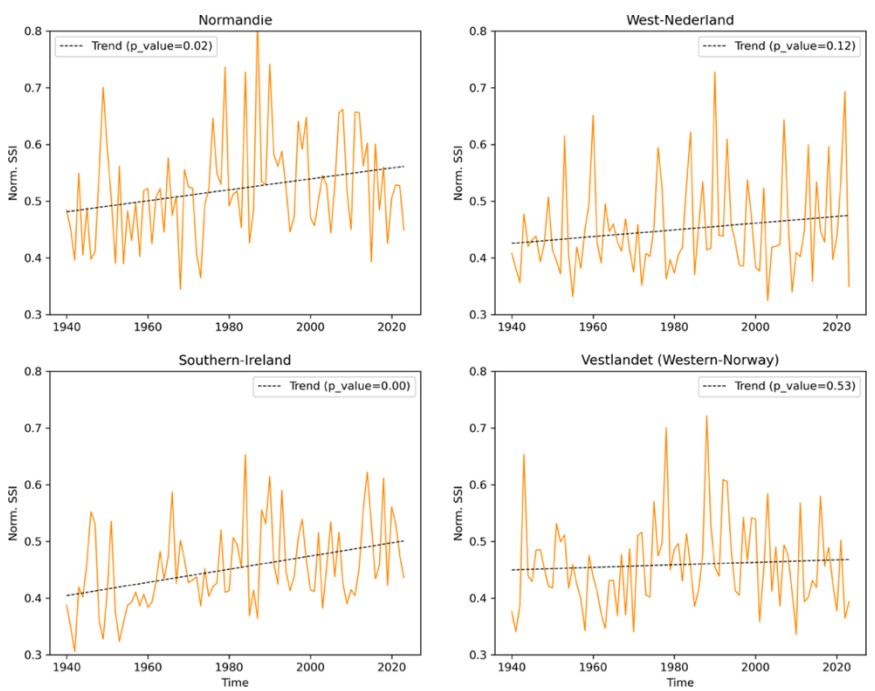

Figure 13. Annual NSSI spatially averaged over four different European regions.

## 5.2 Windstorm applications

In this section, we provide a brief description of the conceptualization and capabilities of the windstorm applications. As anticipated, the first application provides a simple way to visualize the EWS storm tracks on an interactive map over a

selectable period. A single track can then be further explored to inspect summary statistics (minimum MSLP and maximum



10m wind gust speed) or, by clicking on it, to visualize a time series plot of the MSLP and 10m wind gust speed along the track, as well as the underlying storm footprint. The atmospheric variables used to identify the storm tracks are taken from the ERA5 reanalysis and depend on the tracking algorithm. The user-selectable parameters are:

- Tracking algorithm: the algorithm used to identify the storm tracks "TempestExtremes" or "TRACK" (indicated as
"Hodges" to avoid confusion with the term "windstorm track").
- Start date: start date of the time window for which the windstorm tracks are to be fetched from the catalogue and displayed on the map. The first available date is the 1st of January 1940.
- Stop date: stop date of the time window for which the windstorm tracks are to be fetched from the catalogue and displayed on the map. The last available date is the 31st of December 2023. This date will be continuously updated as the catalogue
entry is updated with new ERA5 releases.

The application interface is made of two parts, on the left an input panel and on the right an interactive map, showing the tracks corresponding to the selected date range and tracking algorithm. For a given set of user inputs, the application reads the .csv file containing all the tracks for the selected tracking algorithms and selects the ones within the selected time range. The coordinates contained in the .csv files are converted into line objects and assigned a colour attribute which is proportional to

the MSLP or 10m wind gust along the track in multi-track mode. When a user clicks on a track the application reads the track ID, fetches the corresponding footprint and switches to a single-track modality (Figure 14). In the single-track modality, the MSLP and 10m wind gust speed data are read from the track and plotted in the left-hand panel. The footprint is displayed as a base layer and is available both in its full field and decontaminated versions.

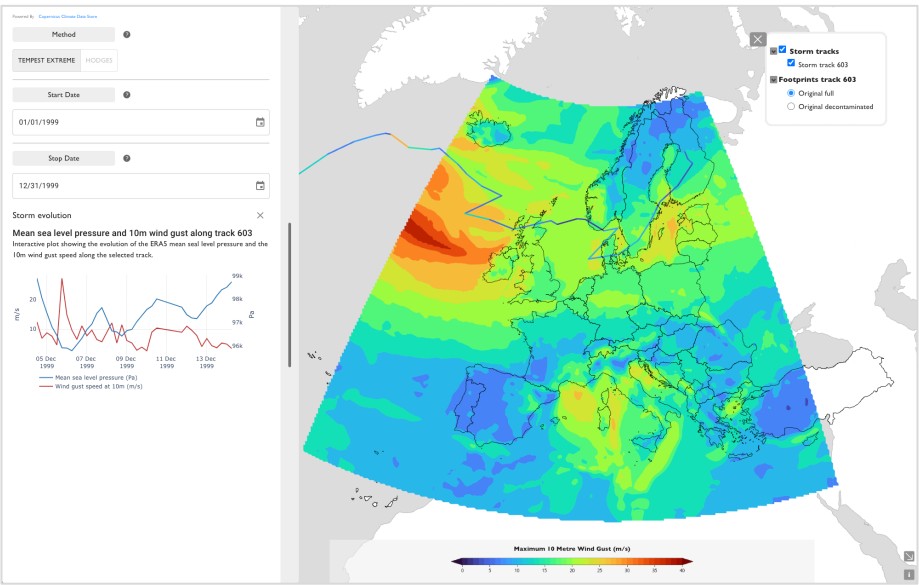

**Figure 14. Overview of the application in single-track modality.**

The second application is based on the above-mentioned four Windstorm Summary Indicators, facilitating the analysis of the

main windstorm tracks and footprint features in terms of the overall event severity.



The application interface is made of three parts (Figure 15a), on the left there is an input panel, at the centre an interactive map showing the regional values of the indicators, and on the right (if a region has been clicked) a time series plot showing the yearly evolution of the selected indicator in the clicked region. Similarly, as in the first application, the user-selectable parameters allow the selection of a specific algorithm, aggregation period or threshold. The interactive map provides a quick

way to qualitatively and quantitatively evaluate the frequency and intensity of windstorms over Europe for a selected period. The interactive map displays a map of Europe showing the yearly value of the selected indicator average over the user-selected year range (Figure 15b). The colour associated with a region is proportional to the value of the indicator, and the scale of colours is provided in the legend at the bottom of the map. To read the exact value for a given region you can hover over the region, and a pop-up window will appear showing the name of the region and the indicator value.

a)

b)

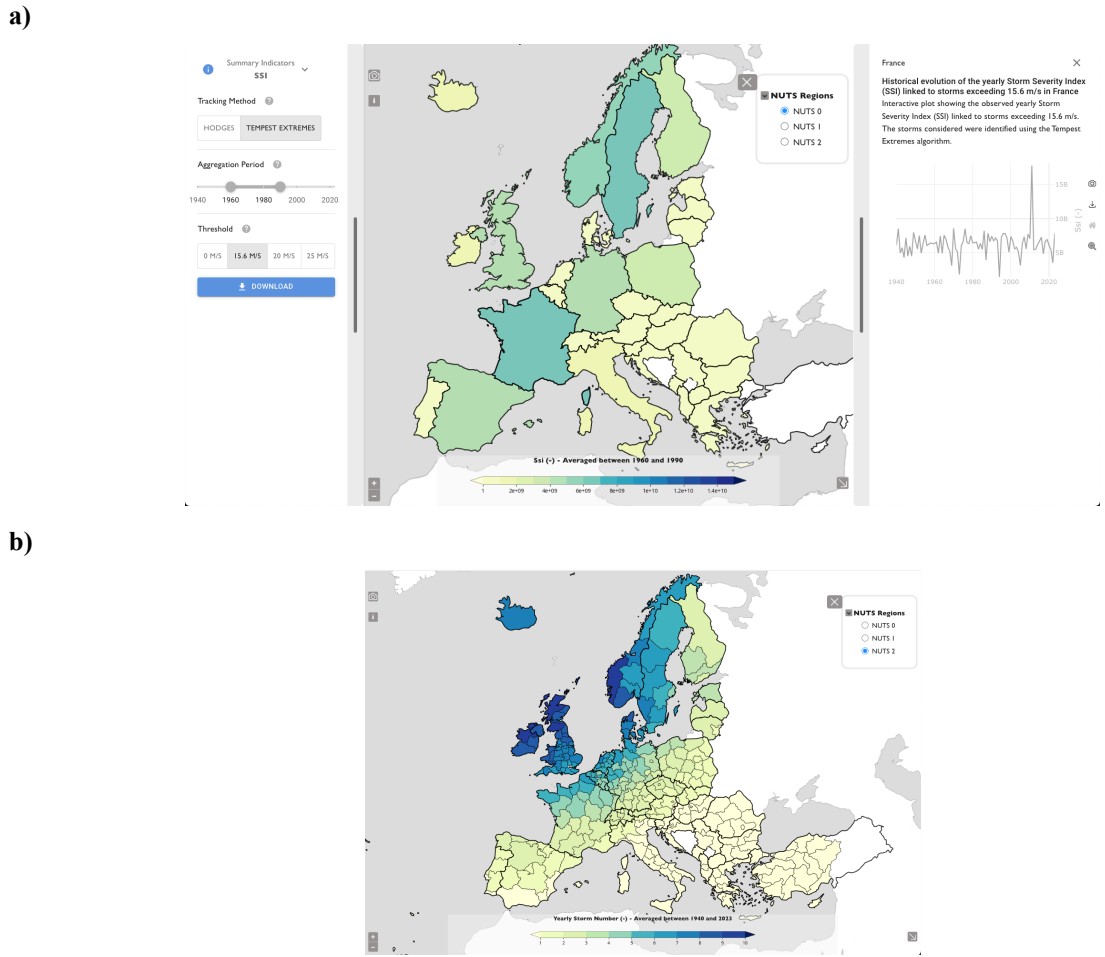

**Figure 15. (a) Application interface with input panel (left), interactive map (centre) and time series panel (right). (b) Interactive map in multi-track mode.**



## 6 Conclusions

A novel extratropical windstorm dataset, EWS, is described, which represents an enhancement and temporal extension of the current C3S Windstorm Service dataset. EWS is composed of two main datasets, based on the ERA5 reanalysis, spanning from 1940 to the present, in a quasi-operational fashion, meaning that the storm tracking algorithm will be automatically run over the new ERA5 reanalyses releases, continuously extending EWS in time. The two datasets consist of windstorm tracks, generated using two different algorithms, TRACK and TempestExtremes (TE) and of the associated spatial footprints. These latter are computed initially from ERA5 data at the original horizontal resolution (approx. 30 km), but also after the application of a statistical downscaling algorithm, based on a multiple linear regression model with a destination horizontal resolution of 1km. Moreover, a set of four summary indicators is further derived, considering three progressively higher wind gust thresholds temporally averaged on an annual basis and spatially averaged over different European regions (NUTS). Windstorm datasets and summary indicators can be explored, and their fitness for the user's purposes assessed, employing two windstorm applications: the first visualizes individual events or aggregation of events and the second allows a more accurate integrated assessment through several windstorm summary indicators.

The present study also provides a first reproducibility assessment of the data, consisting in the evaluation of:

- The capability of the two tracking algorithms to diagnose a set of relevant European windstorms.

- The plausibility of the ERA5 wind gusts footprints derived from the original ERA5 resolution and from a statistically downscaled version of the same data, down to a 1x1 km grid.

- The trends in time of the windstorms MSLP, spatial density and associated wind gust footprints.

Despite the reference dataset limitations and a non-neutral evaluation setting, the two trackers find a notably different number of events when compared to the reference dataset XWS (eXtreme WindStorms), i.e. 40 for TRACK and 18 for TE. It should be noted that all 18 TE tracks are contained in the 40 TRACK tracks. Nevertheless, this analysis does not aim to represent a comprehensive assessment of the quality of the two tracking algorithms, which lies beyond the scope of the present study. This is for two main reasons, firstly for the non-neutral evaluation setting (XWS is based on TRACK) and, secondly, for the temporal/spatial filtering applied to the raw tracks that could unevenly affect the two tracking algorithms in the reproduction of the XWS reference events.

The wind gust footprints associated with the windstorms have been evaluated considering a set of quality-controlled sub-daily observations extracted from the HadISD dataset. In general, observed footprint spatial patterns and wind gust magnitude are well reproduced by ERA5 data, especially over the UK and northwest France. However, specifically over the Netherlands, ERA5 seems to overestimate wind gust maxima footprints for several storms. The statistical downscaling applied to the ERA5 variables generally improves wind gust representation, especially over the right tail of the distribution, representing better the most severe wind gust values. Despite the above-mentioned limitations, in grid-point vs. observation-point evaluation, ERA5 generally represents well footprint spatial patterns and wind gust magnitude. Nevertheless, the statistical downscaling





procedure is shown to be a computationally cheap, viable methodology, complementing the original-resolution ERA5 product, for the definition of crucial information for present and future windstorm-related climate risk analyses.

Regarding the analysis of long-term trends, we found an increasing trend in the density of the windstorm passages over a non-negligible part of the North Atlantic. However, it should be remembered that such result was of sizeable magnitude and statistically significant only using the TRACK algorithm, which in general produces a higher number of detected events over the same climatological period. This sheds some light on the relevance of the choice of the tracking algorithm in such evaluations, as already mentioned in several studies (e.g., Bourdin et al., 2022; Hewson and Neu, 2015; Neu et al., 2013).

Concerning windstorm footprints, statistically significant positive long-term trends are found over part of central and northern Europe during the cold semester, while the two semesters show opposite-sign trends over the British Isles, where a statistically significant increase during the cold semester and a statistically significant decrease during the warm semester could be observed. More generally, the cold semester shows a remarkable and widespread significant increase over Central and Northern Europe, whereas the warm semester shows more diversified, less pronounced trends, with a significant decrease only over the

Central British Isles and a significant increase over Sweden. Concerning the impact introduced by the statistical downscaling, trends spatial patterns and their statistical significance are preserved in moving from the original to the downscaled ERA5, even though a general decrease of the trend magnitude can be observed in the downscaled footprints.

In conclusion, in this study we introduce, and preliminarily evaluate, a new, Enhanced, "quasi-operational" Pan-European Windstorms (EWS) dataset leveraging two well-known storm-tracking algorithms and deriving associated wind gust footprints

at the original ERA5 resolution (30 km approximately) and, after statistical downscaling, down to 1km. Taking advantage of the whole ERA5 time series, EWS datasets present an unprecedented extension, in terms of resolution, time period covered, as well as enabling user/application-defined access, as compared to existing products. These advances are of paramount importance for:

- characterizing the large year-to-year fluctuations in terms of number and intensity of storm passages and associated wind

gust intensity, thus representing a valuable tool for the insurance and reinsurance sectors;

- providing the basis for statistical inferences about long-term trends involving dynamics of Extra-Tropical Cyclones, their multi-decadal climate variability and associated climate risk;

- using EWS datasets as reference products to evaluate the capability of global and regional climate models to reproduce windstorm features and their temporal evolution.


From a user perspective, TRACK captures a more extensive set of events, although these often do not overlap with those identified by the TE algorithm. For this reason, we advocate combining both tracking algorithms to maximise the characterisation of windstorm events. Finally, both the algorithms are also well-suited for the detection of global warming signatures on windstorm features on both global and regional domains, in climate projections as well as for detecting anomalies

on shorter temporal horizons, such as those in seasonal climate predictions.




## 7 Data availability

EWS dataset publication is underway, and it will be publicly available at the Copernicus Climate Change Service (https://climate.copernicus.eu). EWS dataset subset analysed in this study, is publicly available online through Zenodo at https://zenodo.org/records/14554736 (Sangelantoni et al., 2024) and is intended for review processes only. The ERA5
reanalysis data used as an input source for our EWS datasets are publicly available online through the Copernicus Climate Change Service (https://cds.climate.copernicus.eu/). Reference observational sites used for evaluating windstorm footprint wind gust are available at: https://www.metoffice.gov.uk/hadobs/hadisd/. Codes for generating the analyses presented in this study are available from Zenodo at https://zenodo.org/records/14554619 (Sangelantoni, 2024).

*Author contributions.* LS, ST, LC, ES conceived EWS datasets. LS performed the data analysis and produced related codes. PLV and KH provided guidance on the results interpretation and methodological aspects. VM and MA produced IT infrastructure and codes underpinning EWS datasets, windstorm indicators and applications. SA and CC supervised the production of the EWS datasets. All authors discussed the results and contributed to drafting and reviewing the manuscript.

*Competing interests.* The authors declare that they have no conflicts of interest.

*Disclaimer.* EWS dataset publication is underway, the related subset analysed in the present study and available online through Zenodo at https://zenodo.org/records/14554736  is intended for the review process only.

*Acknowledgements.* LS, ST, LC, ES, VM, and MA acknowledge funding from ECMWF-implemented C3S2_413 contract.

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
