# Peer review of "A novel European windstorm dataset based on ERA5 reanalysis from 1940 to present"

_EGUsphere, 2024_

## Referee Comment (RC1)

Thank you for your work in developing the European windstorm dataset based on ERA5 reanalysis.

This work lacks a good motivation for their study as providing windstorm track data from already existing ERA5 reanalysis data for insurance and risk management industry is not an innovation. Not sure why authors did not come up with strong objectives that should result in a journal paper. The authors mentioned, "The objective of this innovation is to promote a knowledge-based assessment of the nature ...," which clearly is vague. The overall paper is written inadequately and hard to follow the style.

In line 55, authors used words, such as "innovation" which I would rather avoid using such words or phrasing without claiming any notable innovation.

The conclusion section is more geared towards "summary and conclusions." Please take care of it.

Authors said, "The choice of the tracking algorithm is shown to be an important factor in the decision-making process, as it results in non-negligible uncertainties in main windstorm statistics," I would suggest that adding a quantifiable result that can really show if it is an important factor or not.

The font sizes of axes, ticks, titles, captions, etc. are non-uniform in most of the figures. Same goes for colorbar as well. Please be sure to make them uniform. The figure resolution needs to be enhanced for better readability as they appear to be of low resolution in the current version of the manuscript. In addition, some of the figure captions are inadequately written without adequate sub-plot numbers, such as a, b, c, etc. for the reader. Make sure to provide numbering to all sub-plots and be consistent with the results and discussions provided.

---

## Author Comment (AC1)

**A novel European windstorm dataset based on ERA5 reanalysis from 1940 to present**

Lorenzo Sangelantoni[1], Stefano Tibaldi[1], Leone Cavicchia[1], Enrico Scoccimarro[1], Pier Luigi Vidale[2], Kevin I. Hodges[2], Vivien Mavel[3], Mattia Almansi[3], Chiara Cagnazzo[4], and Samuel Almond[4]

[1] CMCC Foundation - Euro-Mediterranean Center on Climate Change, Bologna, Italy

[2] National Centre for Atmospheric Science, Dept. of Meteorology, University of Reading, Reading, UK

[3] B-Open Solutions srl, Rome, Italy

[4] ECMWF, Bonn, Germany

Correspondence to: Lorenzo Sangelantoni (lorenzo.sangelantoni@cmcc.it)

**Response to Reviewer #2**

The authors present an overview of the latest C3S windstorm information service. While I appreciate that this service is useful and presents a fantastic resource for the users highlighted in the manuscript, I have many issues with the presentation of the manuscript in its current form and do not believe it can be published as is. I believe the writing is quite lazy and overly long in places, with necessary details being left out. Below i detail my major comments, and also minor points. Once these are addressed, I think the authors should resubmit to the journal for another consideration.

We would like to thank anonymous Reviewer #2 (hereafter Rev2) for the detailed and constructive comments provided, invaluable in guiding the revision and reorientation of the manuscript.

Integrating the comments received from the two Reviewers, we propose a revised version of the manuscript, focusing exclusively on the analysis of extratropical cyclone (ETC) diagnostics derived from the ETC track datasets. Other components of the windstorm service included in the original version will be succinctly introduced but not examined to sharpen the scientific focus and make the manuscript more concise.

The core of the revised manuscript will be represented by a set of twelve ETC diagnostics (Table R2.1), characterizing dynamical and impact-relevant features of ETCs like wind gusts and precipitation (Corner et al., 2025). Here, we will examine the modulation introduced by considering ETCs detected by two tracking algorithms leveraging two different variables to identify cyclone centers (850hPa relative vorticity and mean sea level pressure (mslp) for TRACK and TempestExtremes (TE) respectively).

Analysis based on ETC diagnostics enables the following: (i) to perform a thorough examination of the structural differences of the ETC detected by the two tracking algorithms used in the

windstorm service, and (ii) to explore whether ERA5 can be reliably used for trend detection of ETC diagnostics and its applicability in climate impact studies. In this regard, following suggested studies (Bloomfield et al., 2018; Cusack, 2023; Wohland et al., 2019) and (Scoccimarro et al., 2024) we examine how the selected time period may influence the sign and robustness of possibly detected trends considering the evolving nature of ERA5's data assimilation system over decades. Within the set of ETC diagnostics, special attention will be paid to precipitation-based ETC diagnostics, where ERA5 trends will be compared against other observational reference datasets such as MSWEP (Beck et al., 2019) and whether changes in precipitation extremes associated with ETCs are consistent with expectations from the Clausius–Clapeyron relationship (Pall et al., 2007; Scoccimarro et al., 2024).

The revised version will be built on a more defined methodological framework, incorporating the main comments raised by the Reviewers and articulating as follows:

(i) the characterization of the mean features of the ETCs detected by the TRACK and TE algorithms;

(ii) ETC diagnostics trends across different periods;

(iii) a comparison of ETC precipitation diagnostics with reference datasets and physical scaling expectations;

(iv) a discussion on the implications of the analyzed periods on ETC diagnostic trends detection.

After having outlined the main pillars of the revised manuscript, we then provide a point-by-point response to Rev2 comments (in blue), highlighting the parts that we believe should be included in the revised manuscript.

**Major comments**

(i) My biggest comment involves section 4 on the trend quantification. I believe that these results are highly inrepresentative of changes to windstorms across Europe, and while the authors present significant trends, consideration should be made as to whether to even include this analysis. It has been shown that trends from long reanalysis products are highly questionable (e.g. Wohland et al., 2019; Bloomfield et al., 2018; https://agupubs.onlinelibrary.wiley.com/doi/full/10.1029/2018JD030083; https://iopscience.iop.org/article/10.1088/1748-9326/aad5c5), so this poses as to whether the windstorms in the early part of the ERA5 catalogue are representative of true climate. Furthermore, variability in European windstorms is highly non-linear and dominated by peaks in the 90s (see Cusack, 2023, fig 9; https://nhess.copernicus.org/articles/23/2841/2023/) and therefore the authors application of a linear trend is not appropriate. I believe the authors should not use this data to quantify any potential trends but instead present this as a resource for analysis in a manner the end-user deems appropriate. This section should be (in my opinion) removed for the resubmission.

We acknowledge Rev2's concern about the robustness of long-term trends based on the whole available period from ERA5 data (1940–2023). To address this aspect, we assess the impact on trends resulting by considering different periods.

Figure R2.1 presents spatial patterns of mean values and linear trends in ETC wind gust footprints derived using both the TRACK and TE algorithms, across two distinct periods: 1940–2023 and 1979–2023. For the full ERA5 period (1940–2023), results reveal a dipole structure, with significant positive trends over much of the North Atlantic and the European continent, and significant negative trends over the central-western Atlantic. The spatial patterns are broadly consistent between the two tracking algorithms. In contrast, for the shorter period (1979–2023), significant trends exhibit a longitudinal displacement, with significant positive trends confined primarily to the western North Atlantic. Furthermore, discrepancies between the outputs of the two tracking algorithms become more pronounced during this latter period.

To further investigate this issue, we expanded our trend analysis beyond wind gust footprints to include a broader set of 12 ETC diagnostics (defined and summarized in Table R2.1), recently proposed by Corner et al. (2025). The first seven diagnostics are computed from core storm properties (e.g., minimum sea-level pressure) provided directly by the tracking algorithms. The remaining five involve additional fields, such as 10m wind speed, 850 hPa relative vorticity, and total precipitation, sampled along each cyclone track and averaged over its lifetime.
Figure R2.2 displays trends for a subset of diagnostics that show significant linear trends (via the Mann–Kendall test) in at least one of the two time periods and from at least one tracking algorithm. Asterisks denote levels of statistical significance: *** $p < 0.001$, ** $0.001 < p < 0.01$, * $0.01 < p < 0.05$.

Results in Figure R2.2 confirm that trend estimates are sensitive to the considered period, particularly for diagnostics based on mslp, 10m wind gusts and relative vorticity. In contrast, trends in precipitation-based diagnostics appear more stable across the two periods in both trends' magnitude and statistical significance. This analysis underlines the importance of interpreting long-term ETC trends in reanalysis data with caution and reinforces our decision not to emphasize long-term trends in the revised manuscript, favoring comparisons of specific periods.

[Figure]

Figure R2.1 Windstorm footprints derived considering TRACK (a) and TE (b) ETC tracks mean values (upper panels) and linear trends (bottom panels) considering two different periods, 1940-2023 (left columns) and 1979-2023 (right columns). Stippling indicates significant trends (p-value < 0.05)

| ETC Diagnostic | Definition |
|---|---|
| Genesis lat (°) | Latitude of the first ETC track time step. |
| Meridional displacement (°) | Difference between the latitude of the first and last ETC track time step. |
| Min mslp lat (°) | Latitude of the lowest ETC track mslp value. |
| Min mslp (hPa) | Lowest ETC track mslp value. |
| Deepening rate/24h (hPa) | Difference between the lowest ETC track mslp and the mslp value 24 hours before (negative=deepening). |
| Lifetime (days) | ETC duration. |
| Mean speed (km/h) | Mean wind speed considering the whole ETC. |
| Max 850hPa rel. vort. (s-1) | Maximum 850hPa relative vorticity considering all the ETC time steps. |
| Max 10m wind gust (m/s) | Maximum 10m wind gust considering all the ETC time steps. |
| Max areal pr (mm/6hr) | Maximum accumulated 6-hourly precipitation associated with each ETC time step and considering an area with 12 deg. Geodesic radius surrounding ETC points (as in Corner et al., 2025). |
| Max point pr (mm/6hr) | Maximum grid-point-specific accumulated 6-hourly precipitation associated with ETC time steps and considering an area with 12 deg. Geodesic radius surrounding ETC points. |
| Sum pr (mm/event) | Cumulative 6-hourly precipitation considering all the ETC time steps on an area with 12 deg. Geodesic radius surrounding ETC points. |

Table R2.1. ETC diagnostics derived considering the original tracking algorithm tracks

[Figure]

Figure R2.2 ETC diagnostic trends (Mann-Kendall test slope) derived over two different periods 1940-2023 and 1979-2023 shown in panels (a) and (b), respectively with TRACK (upper-left triangle) and TE (lower-right triangle) algorithms.

(ii) My other major comment is around the statistical downscaling used to generate the high-resolution footprints. It appears from the equation at lines 208-209 that the only information at the 1km-scale is the local terrain. Therefore, all the other terms are there as re-scalings and then the information to downscale is just an orography scaling. The reason i take issue for this is in situations when you may have something like a downslope windstorm. You would expect the strongest gusts to be at the lowest elevation, yet following this statistical approach the strongest gusts would be at the highest elevations? This is surely unintuitive. What validation of this downscaling has gone on, and why was this chosen over a dynamical downscaling approach that happened in the previous C3S windstorm product WISC?

We thank the reviewer for this comment, which indeed involves a key methodological consideration regarding the statistical downscaling of wind gusts. However, as previously introduced in our response in the revised manuscript, to avoid broadening the scientific focus too much, we will not include the wind footprint dataset and associated statistical downscaling method issues.

As mentioned by Rev2, the downscaling formulation exploits two main sources of information beyond the maximum wind gusts as provided by ERA5: higher-resolution orography (1 km) and the wind shear between two heights. As correctly pointed out, higher-resolution orography is the only predictor introducing the resolution enhancement. However, wind shear, despite being used at the same original ERA5 resolution adjusts wind gusts leveraging information from turbulence theory which estimates wind gusts based on the difference in wind speeds between two heights (10 m and 100 m), and the logarithmic ratio of these heights, leveraging the logarithmic relationship of wind profile characterizing the lower part of the planetary boundary layer. Regarding the comment "You would expect the strongest gusts to be at the lowest elevation, yet following this statistical approach the strongest gusts would be at the highest elevations?" For this specific context, the limitation, beyond the known issues of the downscaling formulation, which, as noted by van den Brink (2019), has not been explicitly designed for high-elevation or complex terrain contexts, should be attributed to the model's inability to represent key physical mechanisms underlying downslope windstorms. Hydrostatic atmospheric models, such as CY41R2 of the ECMWF Integrated Forecast System (IFS), inherently lack the ability to resolve these nonlinear and small-scale processes, filtering out vertical accelerations and thereby limiting the model's capability to simulate the strong vertical motions, wave-breaking, and associated turbulence that are characteristic of downslope windstorms. Furthermore, other essential mechanisms, such as shallow and deep convection, are either absent or only crudely parameterised in such models, further limiting their ability to realistically capture mesoscale flow interactions with complex terrain.

Specifically, about the comment: "What validation of this downscaling has gone on?", downscaling formulation and multiple linear regression model parameters selection are defined and evaluated by van den Brink (2019). In the revised version of the manuscript, we complement

the analysis shown in the first version of the manuscript with the scatterplots shown in Figure R2.3 exemplarily for a subset of six historical well-known events from the reference XWS dataset (Roberts et al., 2014).

Here, we report on the upper-left-hand panels HadISD dataset (Dunn et al., 2016), observational sites providing wind gust variables during the 72-hour time window centred over the most intense phase of the ETC's life. The nearest grid points of original and downscaled ERA5 wind gusts are shown in the upper-central and upper-right panels, respectively. Scatterplots in the bottom panels show original (bottom-left) and downscaled (bottom-right) wind gusts matching observed wind gusts. Acknowledging intrinsic limitations of a point-scale-based evaluation (e.g., sites' local morphological features and representativeness error arising when point-scale and cell-averaged values are compared (Sangelantoni et al., 2019; Gennaretti et al., 2015)), downscaling shows a pattern that consists mainly of a reduction, though not systematic, of the original-resolution underestimation of the right tail wind gusts, and leaving mostly unchanged (lower wind intensity) left tail of the wind gusts distribution. In this regard, we include two specific cases (u19830118 and u19830201) to show that this tendency still holds in cases where the downscaled footprint degrades the original resolution footprint mean bias.

[Figure]

[Figure]

[Figure]

[Figure]

Figure R2.3. The upper left panels show HadISD observational sites providing wind gust variables during the 72-hour time window centred over the most intense phase of the ETC. The nearest grid points of the original and downscaled ERA5 wind gusts are shown in the upper-central and upper-right panels, respectively. Scatterplots in the bottom panels show original (bottom-left) and downscaled (bottom-right) wind gusts matching observed wind gusts. "MB" stands for the mean bias of the ERA5 (original resolution and downscaled) and observed values.

About **"Why was this chosen over a dynamical downscaling approach that happened in the previous C3S windstorm product WISC?"** We fully agree that dynamical downscaling represents a more physically consistent method for capturing high-resolution atmospheric processes, particularly in complex terrains.

However, the choice to adopt a statistical downscaling approach in our workflow was motivated by practical limitations connected with the quasi-real-time nature of the Enhanced Windstorm Service, which requires timely (i.e., on a monthly basis) updates aligned with the regular release of new ERA5 data. Beyond that, extending high-resolution dynamical simulations over the entire multi-decadal period would involve the use of computational resources exceeding availability resources. For these reasons, a statistically based approach was considered the most feasible path for delivering a pan-European, quasi-operational windstorm service while acknowledging inherent trade-offs.

Dynamically downscaled footprint referred to by Rev2 represented a limited proof-of-concept experiment using the HadGEM3 GA3 and GL3 configurations of the Met Office Unified Model (MetUM) at 25 km resolution was included in the previous service, this covered only a subset of the whole period (1985–2011) and was not intended as a full operational component. Moreover, the 25 km resolution is still relatively coarse for processes such as deep convection or

orographically induced downslope windstorms, which typically require km-scale resolution to be realistically represented.

(iii) The choice of the 990 hPa track point threshold is one i question. My opinion is that often frontal wind gusts may occur when the cyclone core pressure is >990 hPa. What is the authors' justification for this, and what impact does this choice have on the footprints that they are creating?

The filtering strategy applied to the original tracks detected by the two tracking algorithms aims at isolating a manageable number of events relevant to the purpose of the datasets, namely serving the insurance sector for mitigating windstorm climate risk. However, we acknowledge the concern of Rev2, such that in the revised version of the manuscript, core analyses will involve the original outputs of the tracking algorithms. Moreover, the windstorm service will also supply non-filtered ETC tracks for advanced users. More specifically, we have performed an analysis of ETC wind gust footprints considering filtered and unfiltered tracks, for the two tracking algorithms, to examine the impact of the filtering strategy put into question. In Figure R2.4, we show unfiltered (a,c) and filtered (b,d) ETC tracks produced by TRACK (a,b) and TE (c,d) algorithms. Each subplot reports ETC tracks wing gust footprint mean values (upper sub-panels) and the associated trends (bottom sub-panels) considering the whole available period. As expected, a mean wind gust is much higher when considering the filtered ETCs, with, however, comparable trends and spatial patterns, especially considering TRACK results, over the northwestern European sector, as already emphasized in the first version of the manuscript. However, as previously mentioned, in the revised paper, we will base our analyses on original (unfiltered) ETC tracks.

[Figure]

Figure R2.4 TRACK (a,b) and TE (c,d) unfiltered (a,c) and filtered (b,d) ETC tracks wind gust footprints. Footprint mean values and associated trends are shown in the upper and lower sub-panels, respectively.

**Minor comments**

L20 - "an increasing". To be corrected accordingly.

Use season instead of semester throughout. The semester has been used to respond to service requirements. Results for the season (e.g., DJF) will be provided in the supplementary material.

L20 "associated footprint wind gusts magnitude" doesn't read well and should be rephrased. This sentence, referring to the trends analysis, will likely be removed in the revised version of the manuscript abstract.

L21 "portion of the European territory" should be more specific. Will be changed into the northwestern sector.

L29 - "During recent decades". THis is not a new phenomena and this makes it seem as if it is. Rephrase. Rephrased: "Europe has long experienced highly impactful windstorms, which in recent decades have continued to cause significant human and economic losses."

L31 - reference for the €5 billion per year. To be adjusted accordingly.

L48 - also consider referencing the recent CMIP6 study (https://rmets.onlinelibrary.wiley.com/doi/10.1002/qj.4849). The reference has been added.

L58 - you use "innovation" here, but other windstorm assessments have been performed in the past. You mention XWS, but also there should be some recognition of WISC as this was the predecessor of this project and provided all this information but instead with ERA-Interim. Perhaps use "resource" instead of "innovation". To be modified accordingly.

L75-77 - there are >>2 tracking schemes, so surely just using these two does not represent the full uncertainty from tracking? See recent paper by Flaounas et al. (https://wcd.copernicus.org/articles/4/639/2023/) as to the tracking uncertainty. If the authors are only using these two, some justification as to why these two is required. We fully acknowledge that using only two tracking algorithms does not capture the full range of uncertainty associated with cyclone identification, as highlighted in the recent work by Flaounas et al. (2023). However, the two algorithms used in our study were selected because they represent two fundamentally different tracking approaches based on distinct variables (i.e., relative vorticity and mslp), most used in ETC and often associated with differing sensitivities to storm intensity and structure. While this selection does not exhaust the diversity of existing tracking schemes, it provides a meaningful contrast between these two widely adopted methodologies, thereby offering a first-order estimate of structural uncertainty. These considerations will be added in the revised version of the manuscript.

L125/126 - you state earlier that you require winds within a 3-degree radius, but only mention 5 degrees here. This is either an additional step that needs mentioning or needs rectifying here. 3-

and 5-degree radii refer to different detection processes. The 3-degree radius refers to the track wind gust footprint detection. The 5-degree threshold is internally used by the TRACK algorithm in the detection of the mslp minima associated with the minima of 850hPa rel. vorticity. This will be further clarified in the revised manuscript.

L144-146 - so these tracks must essentially pass over Europe. Please make this clearer here as this is quite a clunky section and i found it hard to follow these simple regional criteria. Thanks for the suggestion, this will be rephrased and made clearer in the revised manuscript.

L153-154 - this sentence doesn't make the most grammatical sense, please rephrase more simply to just mention TRACK and TE in (a) and (b) respectively. We have redone Figure 1 to make clearer the main features (mean mslp and density) of ETCs detected by the two algorithms (Figure R2.5).

[Figure]

Figure R2.5 mean mslp (upper panels) and ETC passages mean density (bottom panels) as reproduced by TRACK (left) and TE (right) algorithms.

Figure 1 - this would be better as a track density plot and differences. It's very hard to see any differences in the top panels with this density of lines. As above.

L164 - "Moreover, TRACK is quite consistent with itself", what do you mean by this? Of course it is consistent with itself! Figure 2 has been removed since we derived a set of ETC track diagnostics for elucidating the structural difference of ETCs detected by the two tracking algorithms.

Figure 2 - please add a colorbar for the shading. Also, the black contours are not consistent across panels, so it makes it very difficult to compare the values, especially when looking at the panels on the top row. See above.

You need to state in your methods what months are you "cold" and "warm" seasons. Months making up the cold semester are now specified (November to March).

L172 - why are you using a 4x4 degree binning of data for your statistics? This is incredibly coarse and also much coarser than the native ERA5 data. It seems unnecessary and should be done on at least 1x1 degree surely as ERA5 is 0.25x0.25 degree. For the sake of the readability of the plots, a grid of 2.5 x 2.5 degrees has been adopted. This grid has been considered for Figure R2.5.

Figure 4 - it would be good to see a different plot between these two footprints. As the tracks are different, but the gusts are the same i question quite why we need to see both, but it would be good to see how this affects the resultant footprints. In the revised manuscript, we will provide a comparison of the climatological mean and trends of the whole set of footprints associated with the ETC tracks produced by the two algorithms, Figure R2.1.

L235 - please mention dates of these storms. Storm dates are shown in Figure 7 of the original manuscript and will be explicitly mentioned in the text of the revised manuscript.

L237 - 'traces' should be 'tracks'? This is correct, thanks.

Figure 7 - it feels unnecessary to have 9 panels here, can you not plot all the tracks for a storm on one panel but with different colours? Yes, in the revised manuscript, we will produce a three-panel plot reporting all five tracks for each of the three reference events.

L247 - how do you propose someone would combine both algorithms? One possible solution would be to build statistics by considering the ETCs from both tracking algorithms, assigning a higher confidence level to the ETCs detected by both algorithms.

Figures 8/9, i recommend somehow combining figures 8 and 9, or just using figure 9, as this is in some ways duplicate information and means that the manuscript is unnecessarily long. We now propose an event-specific single-plot solution like the ones shown in Figure R2.3.

Page 14 - this is a very long paragraph - consider reducing and breaking up the text throughout the manuscript. This section of the results will be significantly condensed in the revised manuscript.

L279/280 - how much better though? Is it possible to compare the mean bias of the downscaled ERA5 to the original ERA5 relative to the obs for selected (or all) storms. This will demonstrate the benefit of the downscaling. Mean bias has been added in the corresponding plots in Figure R2.3.

L345-350 - this SSI formulation is somewhat different to what is commonly used (see Leckebusch et al., 2008). As SSI is very dependent on the formulation i recommend using the most commonly adopted formulation (e.g. Karremann et al., 2014; https://iopscience.iop.org/article/10.1088/1748-9326/9/12/124016). We acknowledge the sensitivity of the Storm Severity Index (SSI) to its formulation. For this first version of the service, we would stick with the current approach to maintain consistency with the workflow

development. However, we recognise the value of commonly used alternatives and plan to assess SSI sensitivity to different formulations in future service updates.

**References**

Beck, H. E., Wood, E. F., Pan, M., Fisher, C. K., Miralles, D. G., Van Dijk, A. I. J. M., McVicar, T. R., and Adler, R. F.: MSWep v2 Global 3-hourly 0.1° precipitation: Methodology and quantitative assessment, Bull. Am. Meteorol. Soc., 100, 473–500, https://doi.org/10.1175/BAMS-D-17-0138.1, 2019.

Bloomfield, H. C., Shaffrey, L. C., Hodges, K. I., and Vidale, P. L.: A critical assessment of the long-term changes in the wintertime surface Arctic Oscillation and Northern Hemisphere storminess in the ERA20C reanalysis, Environ. Res. Lett., 13, https://doi.org/10.1088/1748-9326/aad5c5, 2018.

van den Brink, H. W. (2020). An effective parametrization of gust profiles during severe wind conditions. Environmental Research Communications. Institute of Physics. https://doi.org/10.1088/2515-7620/ab5777

Cusack, S.: A long record of European windstorm losses and its comparison to standard climate indices, Nat. Hazards Earth Syst. Sci., 23, 2841–2856, https://doi.org/10.5194/nhess-23-2841-2023, 2023.

Dunn, J. H. R., Willett, M. K., Parker, E. D., & Mitchell, L. (2016). Expanding HadISD: Quality-controlled, sub-daily station data from 1931. Geoscientific Instrumentation, Methods and Data Systems, 5(2), 473–491. https://doi.org/10.5194/gi-5-473-2016

Flaounas, E., Aragão, L., Bernini, L., Dafis, S., Doiteau, B., Flocas, H., et al. (2023). A composite approach to produce reference datasets for extratropical cyclone tracks: application to Mediterranean cyclones. Weather and Climate Dynamics, 4(3), 639–661. https://doi.org/10.5194/wcd-4-639-2023

Gennaretti, F., Sangelantoni, L., and Grenier, P.: Toward daily climate scenarios for Canadian Arctic coastal zones with more realistic temperature-precipitation interdependence, J. Geophys. Res. Atmos., 120, 11,862-11,877, https://doi.org/10.1002/2015JD023890, 2015.

Pall, P., Allen, M. R., and Stone, D. A.: Testing the Clausius-Clapeyron constraint on changes in extreme precipitation under CO2 warming, Clim. Dyn., 28, 351–363, https://doi.org/10.1007/s00382-006-0180-2, 2007.

Roberts, J. F., Champion, A. J., Dawkins, L. C., Hodges, K. I., Shaffrey, L. C., Stephenson, D. B., et al. (2014). The XWS open access catalogue of extreme European windstorms from 1979 to 2012.

Natural Hazards and Earth System Sciences, 14(9), 2487–2501. https://doi.org/10.5194/nhess-14-2487-2014

Sangelantoni, L., Russo, A., & Gennaretti, F. (2019). Impact of bias correction and downscaling through quantile mapping on simulated climate change signal: a case study over Central Italy. Theoretical and Applied Climatology, 135(1–2), 725–740. https://doi.org/10.1007/s00704-018-2406-8

Scoccimarro, E., Lanteri, P., and Cavicchia, L.: Freddy: breaking record for tropical cyclone precipitation?, Environ. Res. Lett., 19, https://doi.org/10.1088/1748-9326/ad44b5, 2024.

Wohland, J., Omrani, N. E., Witthaut, D., and Keenlyside, N. S.: Inconsistent Wind Speed Trends in Current Twentieth Century Reanalyses, J. Geophys. Res. Atmos., 124, 1931–1940, https://doi.org/10.1029/2018JD030083, 2019.

---

## Author Comment (AC2)

**A novel European windstorm dataset based on ERA5 reanalysis from 1940 to present**

Lorenzo Sangelantoni[1], Stefano Tibaldi[1], Leone Cavicchia[1], Enrico Scoccimarro[1], Pier Luigi Vidale[2], Kevin I. Hodges[2], Vivien Mavel[3], Mattia Almansi[3], Chiara Cagnazzo[4], and Samuel Almond[4]

[1] CMCC Foundation - Euro-Mediterranean Center on Climate Change, Bologna, Italy

[2] National Centre for Atmospheric Science, Dept. of Meteorology, University of Reading, Reading, UK

[3] B-Open Solutions srl, Rome, Italy

[4] ECMWF, Bonn, Germany

Correspondence to: Lorenzo Sangelantoni (lorenzo.sangelantoni@cmcc.it)

**Response to Reviewer #1**

Thank you for your work in developing the European windstorm dataset based on ERA5 reanalysis.

This work lacks a good motivation for their study as providing windstorm track data from already existing ERA5 reanalysis data for insurance and risk management industry is not an innovation. Not sure why authors did not come up with strong objectives that should result in a journal paper. The authors mentioned, "The objective of this innovation is to promote a knowledge-based assessment of the nature ...," which clearly is vague. The overall paper is written inadequately and hard to follow the style.

We would like to thank Reviewer #1 (hereafter Rev1) for the comments provided. These suggestions were valuable in guiding the revision and reorientation of the manuscript.

Integrating the comments received from the two Reviewers, we propose a revised version of the manuscript, focusing exclusively on the analysis of extratropical cyclone (ETC) diagnostics derived from the ETC track datasets. Other components of the windstorm service included in the original version will be succinctly introduced but not examined to sharpen the scientific focus and make the manuscript more concise.

The core of the revised manuscript will be represented by a set of twelve ETC diagnostics (Table R2.1), characterizing dynamical and impact-relevant features of ETCs like wind gusts and precipitation (Corner et al., 2025). Here, we will examine the modulation introduced by considering ETCs detected by two tracking algorithms leveraging two different variables to identify cyclone centers (850hPa relative vorticity and mean sea level pressure (mslp) for TRACK and TempestExtremes (TE) respectively).

Analysis based on ETC diagnostics enables the following: (i) to perform a thorough examination of the structural differences of the ETC detected by the two tracking algorithms used in the windstorm service, and (ii) to explore whether ERA5 can be reliably used for trend detection of ETC diagnostics and its applicability in climate impact studies. In this regard, following suggested studies (Bloomfield et al., 2018; Cusack, 2023; Wohland et al., 2019) and (Scoccimarro et al., 2024) we examine how the selected time period may influence the sign and robustness of possibly detected trends considering the evolving nature of ERA5's data assimilation system over decades. Within the set of ETC diagnostics, special attention will be paid to precipitation-based ETC diagnostics, where ERA5 trends will be compared against other observational reference datasets such as MSWEP (Beck et al., 2019) and whether changes in precipitation extremes associated with ETCs are consistent with expectations from the Clausius–Clapeyron relationship (Pall et al., 2007; Scoccimarro et al., 2024).

The revised version will be built on a more defined methodological framework, incorporating the main comments raised by the Reviewers and articulating as follows:

(i) the characterization of the mean features of the ETCs detected by the TRACK and TE algorithms;

(ii) ETC diagnostics trends across different periods;

(iii) a comparison of ETC precipitation diagnostics with reference datasets and physical scaling expectations;

(iv) a discussion on the implications of the analyzed periods on ETC diagnostic trends detection.

After having outlined the main pillars of the revised manuscript, we then provide a point-by-point response to Rev1 comments (in blue), highlighting the parts that we believe should be included in the revised manuscript.

In line 55, authors used words, such as "innovation" which I would rather avoid using such words or phrasing without claiming any notable innovation. We agree that the use of the term "innovation" may have overstated the contribution in its current form. In the revised manuscript, we will remove or rephrase such terms to more accurately reflect the scope of the work and its contribution, focusing instead on the added value of the service in terms of long-term consistency, methodological transparency, and user relevance.

The conclusion section is more geared towards "summary and conclusions." Please take care of it. The conclusion section will be written accordingly to highlight the main take-home messages of the windstorm service and the scientific results of the paper.

Authors said, "The choice of the tracking algorithm is shown to be an important factor in the decision-making process, as it results in non-negligible uncertainties in main windstorm statistics," I would suggest that adding a quantifiable result that can really show if it is an important factor or not. As previously mentioned, to improve the comparison between the two tracking algorithms, we introduce a set of ETC diagnostics aimed at capturing the main structural

and dynamic features of the identified thunderstorms. Beyond simply counting storms and paths, this approach allows us to identify systematic differences in how intensity and dynamic characteristics compare in the events detected by the two tracking algorithms, and to assess their implications for downstream applications, such as wind footprint estimation. Diagnostics results for the two tracking algorithms, spatially averaged over the whole domain considered and for two different periods are shown in Figure R1.1. Related trends are shown in Figure R2.2.

a

ETC diagnostics PDF (1940-2023)

[Figure]

b

[Figure]

Figure R1.1. Vertically oriented PDFs built on spatially averaged ETC diagnostics considering tracks produced by TRACK and TE algorithms. Two periods are considered: 1940-2023 (a)

and 1979-2023 (b). KS and MW identify the presence and level of statistical significance of the Kolmogorov-Smirnov (KS) and Mann-Whitney U (MW) tests, respectively. Inside each violin plot, the horizontal solid line represents the PDF median, whereas dashed lines the 25th and 75th percentiles.

Figure R1.1 presents the probability density functions (PDFs) of a set of spatially averaged cyclone diagnostics (as defined in Table R2.1), calculated from the tracks identified by the two tracking algorithms (indicated along the x-axis) over two time periods: 1940–2023 (panel a) and 1979–2023 (panel b). Statistical differences between the distributions were assessed using two non-parametric tests: the Kolmogorov–Smirnov (KS) test, which is sensitive to differences in the full shape of the distributions, and the Mann–Whitney U (MW) test, which emphasises differences in central tendency (i.e., the medians). Significance levels are indicated as follows: *** $p < 0.001$, ** $0.001 < p < 0.01$, * $0.01 < p < 0.05$.

Results indicate that the two tracking algorithms detect ETCs with significantly different features. TE algorithm tends to detect fewer (see also event density in Figure R2.5) but more intense ETC. TE-tracked storms are generally characterised by lower minimum sea-level pressure, faster-deepening rates, longer lifetimes, and greater cumulative precipitation. Spatially, TE events exhibit a larger meridional displacement, tend to originate further north and show lower mean translation speeds, pointing to greater persistence, as confirmed by their significantly longer duration. Importantly, these diagnostic distributions are consistent across the two periods considered, with no significant differences observed between the 1940–2023 and 1979–2023 PDFs. This temporal robustness stands in contrast to the trend analyses of the same diagnostics (see Figure R2.2), where substantial differences emerged across periods.

Possible explanations for differences, e.g. the propagation speed difference, lay on the fact that TE relies on pressure minima means and it possibly won't identify systems in strong background flows as the systems won't necessarily have a closed circulation. These systems will be travelling very fast, so TE probably only identifies them after they have undergone development and slowed down, whereas using vorticity allows these fast-moving early stages to be identified (Sinclair, 1997; Sinclair Mark R., 1994). The larger number of systems identified by TE at high latitudes is possibly due to the converging meridians when using a lat-long projection and might also explain why there are deeper systems due to the lower background pressures at higher latitudes. The other issue is that spectral filtering is used by TRACK to focus on synoptic scale cyclones whereas TC does not filter.

The font sizes of axes, ticks, titles, captions, etc. are non-uniform in most of the figures. The same goes for the colorbar as well. Please be sure to make them uniform. The figure resolution needs to be enhanced for better readability as they appear to be of low resolution in the current version of the manuscript. In addition, some of the figure captions are inadequately written without adequate sub- plot numbers, such as a, b, c, etc. for the reader. Make sure to provide numbering to all sub-plots and be consistent with the results and discussions provided. All the Figures of the manuscript have been modified and/or redone with higher resolution (300 DPI).

**References**

Beck, H. E., Wood, E. F., Pan, M., Fisher, C. K., Miralles, D. G., Van Dijk, A. I. J. M., McVicar, T. R., and Adler, R. F.: MSWep v2 Global 3-hourly 0.1° precipitation: Methodology and quantitative assessment, Bull. Am. Meteorol. Soc., 100, 473–500, https://doi.org/10.1175/BAMS-D-17-0138.1, 2019.

Bloomfield, H. C., Shaffrey, L. C., Hodges, K. I., and Vidale, P. L.: A critical assessment of the long-term changes in the wintertime surface Arctic Oscillation and Northern Hemisphere storminess in the ERA20C reanalysis, Environ. Res. Lett., 13, https://doi.org/10.1088/1748-9326/aad5c5, 2018.

Cusack, S.: A long record of European windstorm losses and its comparison to standard climate indices, Nat. Hazards Earth Syst. Sci., 23, 2841–2856, https://doi.org/10.5194/nhess-23-2841-2023, 2023.

Pall, P., Allen, M. R., and Stone, D. A.: Testing the Clausius-Clapeyron constraint on changes in extreme precipitation under CO2 warming, Clim. Dyn., 28, 351–363, https://doi.org/10.1007/s00382-006-0180-2, 2007.

Scoccimarro, E., Lanteri, P., and Cavicchia, L.: Freddy: breaking record for tropical cyclone precipitation?, Environ. Res. Lett., 19, https://doi.org/10.1088/1748-9326/ad44b5, 2024.

Sinclair, M. R.: Objective Identification of Cyclones and Their Circulation Intensity, and Climatology, Weather Forecast., 12, 1997.

Sinclair Mark R.: An objective cyclone climatology for the southern emisphere, Mon. Weather Rev., 122, 1994.

Wohland, J., Omrani, N. E., Witthaut, D., and Keenlyside, N. S.: Inconsistent Wind Speed Trends in Current Twentieth Century Reanalyses, J. Geophys. Res. Atmos., 124, 1931–1940, https://doi.org/10.1029/2018JD030083, 2019.